# Establishment and maintenance of heritable chromatin structure during early *Drosophila* embryogenesis

**Shelby A Blythe\*, Eric F Wieschaus\***

Howard Hughes Medical Institute, Princeton University, Princeton, United States

**Abstract** During embryogenesis, the initial chromatin state is established during a period of rapid proliferative activity. We have measured with 3-min time resolution how heritable patterns of chromatin structure are initially established and maintained during the midblastula transition (MBT). We find that regions of accessibility are established sequentially, where enhancers are opened in advance of promoters and insulators. These open states are stably maintained in highly condensed mitotic chromatin to ensure faithful inheritance of prior accessibility status across cell divisions. The temporal progression of establishment is controlled by the biological timers that control the onset of the MBT. In general, acquisition of promoter accessibility is controlled by the biological timer that measures the nucleo-cytoplasmic (N:C) ratio, whereas timing of enhancer accessibility is regulated independently of the N:C ratio. These different timing classes each associate with binding sites for two transcription factors, GAGA-factor and Zelda, previously implicated in controlling chromatin accessibility at ZGA.

**\*For correspondence:** sblythe@ princeton.edu (SAB); efw@ princeton.edu (EFW)

**Competing interests:** The authors declare that no competing interests exist.

## Introduction

Early during development, the genetic loci involved in cell fate specification and differentiation adopt chromatin structure that allows cells to respond accurately to spatial and temporal developmental cues (*Guenther et al., 2007*; *Zeitlinger et al., 2007*; *Vastenhouw et al., 2010*; *Levine, 2011*; *Li et al., 2014*; *Zhang et al., 2014*; *Harrison and Eisen, 2015*; *Sun et al., 2015*). Cells of the early embryo must propagate these chromatin states following mitosis and DNA replication (*Ferraro et al., 2016*). However, it is unclear how such patterns of chromatin accessibility are established in the short interphases and rapid cell cycle progression that characterize early embryonic development, and it is equally unclear how such patterns are inherited following each cell division.

In order for chromatin structure to be maintained continually over multiple cell divisions, mechanisms must exist to overcome nucleosome disruption during DNA replication and mitosis. Nucleosomes are displaced during replication of the parental DNA strand, and re-establishment of prior open and accessible states requires the direct competition between mechanisms for de novo nucleosome assembly and cis-regulatory factors (*Ramachandran and Henikoff, 2016*). Similarly, mitosis is characterized by highly condensed chromatin structure, and it remains unclear the degree to which prior accessibility states are maintained under these conditions. Notably, despite this highly condensed state, it is possible that the underlying structure retains the overall organization and accessibility state of interphase nuclei (*Earnshaw and Laemmli, 1983*). Indeed, a recent study of chromatin structure in murine erythroblasts revealed that the patterns of DNase hypersensitivity are largely preserved during mitosis (*Hsiung et al., 2015*). We wanted to assess when such features of chromatin structure are acquired during early embryogenesis.

Historically, these questions have been difficult to address in embryos. Classical methodologies for measuring chromatin accessibility (e.g., DNase hypersensitivity) and nucleosome positioning (e.

g., Micrococcal nuclease digestion) typically require large quantities of input chromatin, on the order of $10^3$–$10^5$ embryos. Such a high requirement for input material significantly limits the temporal resolution of these experiments, requiring embryos to be pooled from collections spanning hours. Yet, changes in early embryonic chromatin structure are predicted to occur over minute timescales (*Blythe and Wieschaus, 2015b*). A recently developed approach, ATAC-seq (*Buenrostro et al., 2013*), overcomes many of these limitations, allowing for sensitive measurement of both chromatin accessibility and nucleosome positioning in samples as small as single cells.

ATAC-seq generates profiles of chromatin accessibility by fragmenting the genome with Tn5 transposase. Fragmentation preferentially occurs at open, nucleosome-free regions. Accessible regions of the genome are recovered as small (50–100 bp) fragments. In addition, because nucleosome occupancy is refractory to fragmentation, a second population of larger protected fragments is also recovered, and these are used to predict nucleosome positions (*Buenrostro et al., 2013*). In practice, nucleosome positioning is best estimated by coverage of large protected fragments that are flanked by small accessible fragments (*Schep et al., 2015*). Therefore, the organization of generally 'open' chromatin, composed of a mixture of short tracts of accessible DNA with interspersed nucleosomes is effectively measured by this technique.

In the following, we have applied ATAC-seq to measure changes in chromatin accessibility and nucleosome positioning over multiple cell divisions with 3-min time resolution during the three syncytial cell cycles (NC11-13) preceding the *Drosophila* midblastula transition (MBT) and large-scale zygotic genome activation (ZGA) (*Farrell and O'Farrell, 2014*; *Blythe and Wieschaus, 2015a*; *Harrison and Eisen, 2015*). During this period, nuclei rapidly cycle synchronously between S-phase and M-phase with stereotypic cell cycle timing and little or no distinction between early- and late-replicating chromatin compartments (*Farrell and O'Farrell, 2014*). Despite this intense cell cycle activity, embryos arrive at the MBT having established the initial chromatin state upon which developmental patterning systems will initially operate (*Blythe and Wieschaus, 2015a*; *Harrison and Eisen, 2015*). It is unclear whether such regions are established anew during each cell cycle, or instead whether establishment entails the acquisition of mechanisms to counteract the otherwise deleterious consequences of mitosis and DNA replication. To address this question, we measured how patterns of chromatin accessibility are established and maintained in early embryos.

## Results and discussion

We performed ATAC-seq on samples from 13 timepoints (n $\geq$ 3 embryos per timepoint) spanning NC11 and NC13 in 3-min intervals and called peaks in order to identify when regions of 'open' chromatin first become accessible (see Materials and methods). Within a total set of 9824 accessible peaks, two general temporal classes emerge from this analysis that reflect the timing of the initial acquisition of accessibility: peaks that are accessible early and that persist throughout the entire period of observation (*Figure 1A*, middle panel, and *Figure 1B* 'Open by NC11', n = 3084 (31%)), and peaks that gain accessibility during this period (*Figure 1A*: right panel; and *Figure 1B*: New at NC12 or NC13, n = 6740 (69%)). These peaks were assigned to genomic features based on existing annotations (promoters, insulators, and enhancers; see *Figure 1B* and Materials and methods). During this period, 78% (3027 of 3887) of all promoters and insulators dynamically gain accessibility during NC12 and NC13, whereas 47% (1538 of 3260) of all enhancers are open early, already by NC11. We also detect accessibility at 717 peaks overlapping with 854 experimentally validated enhancers from the Vienna Fly Enhancers collection (19.9% of all enhancers in the collection, n = 3604) (*Kvon et al., 2014*). This subset of enhancers is similarly enriched for sites that gain accessibility by NC11 (41%, 294 of 717: p=1.1x$10^{-7}$ by Fisher's Exact Test for $\pm$ NC11 by $\pm$ Fly Enhancer). These results suggest that, on average, enhancer accessibility precedes that of promoters. This effect is evident on both short and long timescales. Over short timescales on the order of minutes, we observe stable enhancer accessibility precedes that of associated promoters (e.g. *hunchback P1*, *tribbles*, *empty spiracles, Ultrabithorax, sloppy paired 2, forkhead*) (*Figure 1A* and *Figure 1—figure supplements 4–8*). This observation is consistent with a model where promoter accessibility is limiting for activation of expression of these genes in spatially restricted patterns, and that enhancer elements could be 'primed' independently of these promoters in advance of expression. In support of this, we also observe that within the set of Vienna Fly Enhancers overlapping our set of accessible peaks, although 386 (45.2%) of these enhancers drive active gene expression in early embryos (stage 4–6),

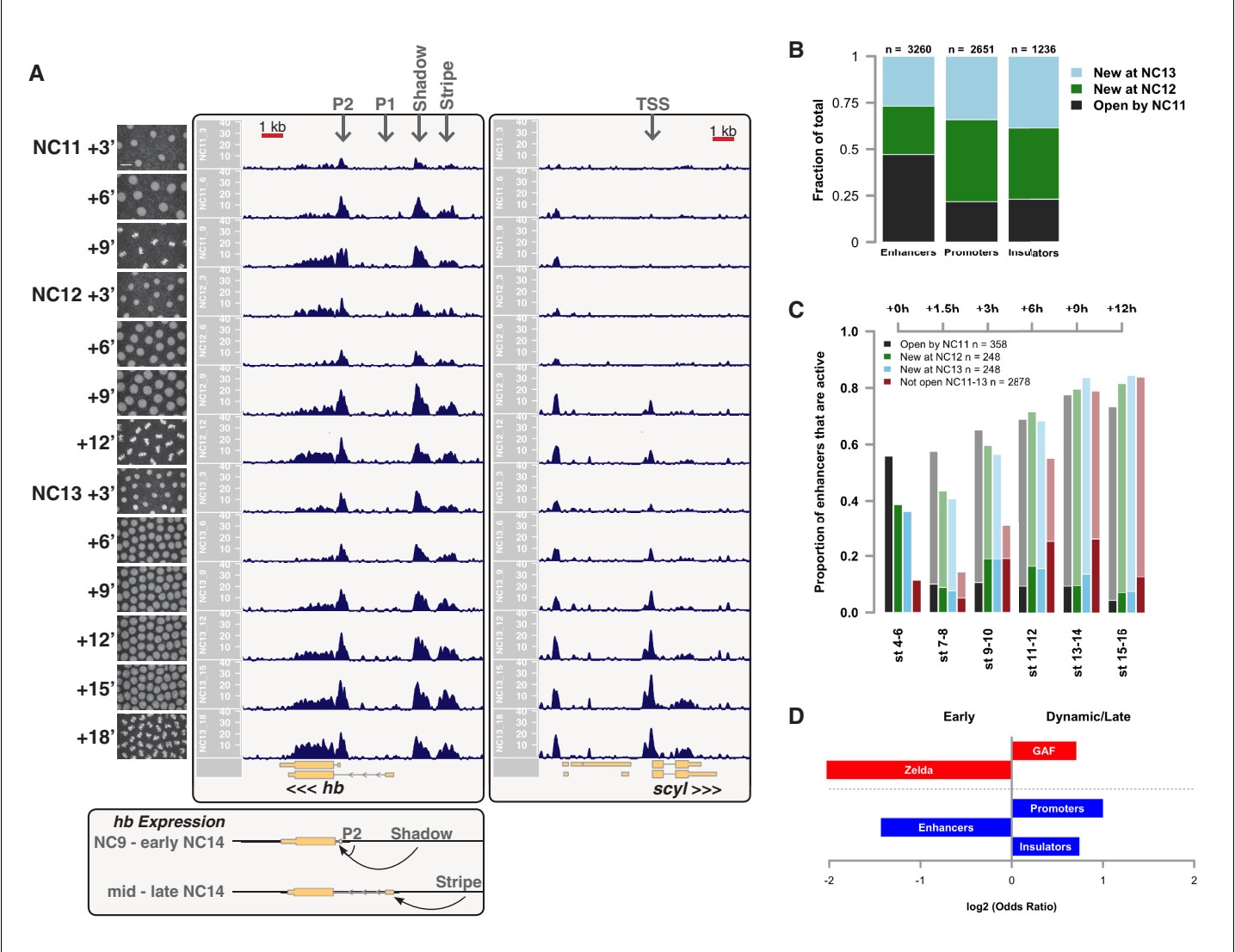

**Figure 1.** Sequential establishment of chromatin accessibility at the MBT. (**A**) (Left column) Single embryos were collected for ATAC seq at the indicated time points. Micrographs show stage-matched panels from a time-lapse image of a single Histone H2Av-GFP embryo. Scale bar = 10 µm. Coverage of sequence reads from accessible chromatin over the *hunchback* (center) and *scylla* (right) loci are shown. The *hunchback* P2 promoter/ enhancer and associated shadow enhancer in addition to the later-acting stripe enhancer (arrows) are open and accessible at all time points measured. A schematic summarizing the regulatory interactions of the *hunchback* locus is shown at bottom left. The *scylla* TSS gains accessibility at NC12 + 9′ and is stably maintained thereafter. Scale bars (red) equal 1 kb. Plots show mean coverage from at least n = 3 replicates. (**B**) The fraction of each genomic feature present during each cell cycle is plotted as a stacked bar chart. 'NC12 new' and 'NC13 new' indicate the set of peaks newly called present in each respective cell cycle. Not shown: 2898 peaks not found to overlap with available genomic annotations used to classify enhancers, promoters, or insulators. (**C**) The association of functionally validated enhancers within each ATAC-seq timing class was calculated, and this plot shows what fraction of these are active at the indicated timepoints. Solid bars indicate the fraction of enhancers whose expression is first detected at the indicated timepoint. The lighter remaining bar indicates the total fraction of active enhancers associated with each ATAC-seq timing class. Color coding is as for panel B, and enhancers not overlapping with the ATAC-seq peaks are shown in red. The estimated elapsed time post NC11-NC13 for each scored stage of development (bottom axis) is indicated on the top axis. (**D**) Odds ratios for enrichment of the indicated genomic features and transcriptional regulators were calculated. Early chromatin accessibility is enriched for enhancers (p=2.54x10$^{-109}$) and binding of Zelda (p=3.7x10$^{-225}$). Late or dynamic chromatin accessibility is enriched for promoters (p=4.62x10$^{-42}$), insulators (p=7.71x10$^{-14}$), and binding of GAF (p=3.76x10$^{-19}$). p-Values are from two-sided Fisher's exact test on contingency tables constructed on [-/+ feature by early/dynamic].

The following figure supplements are available for figure 1:

**Figure supplement 1.** Selection of metaphase-staged embryos and inter-replicate reproducibility.

*Figure 1 continued on next page*

*Figure 1 continued*

**Figure supplement 2.** Fraction of accessible peaks during metaphase.

**Figure supplement 3.** Read-length distribution and comparison of interphase and metaphase library preparations.

**Figure supplement 4.** Browser view of *tribbles*.

**Figure supplement 5.** Browser view of *empty spiracles*.

**Figure supplement 6.** Browser view of *Ultrabithorax*.

**Figure supplement 7.** Browser view of *sloppy paired 2*.

**Figure supplement 8.** Browser view of *forkhead*.

**Figure supplement 9.** Browser view of *giant*.

**Figure supplement 10.** Browser view of *Krüppel*.

**Figure supplement 11.** Browser view of *knirps*.

**Figure supplement 12.** Browser view of *even-skipped*.

**Figure supplement 13.** Browser view of *hairy*.

**Figure supplement 14.** Browser view of *odd-skipped*.

**Figure supplement 15.** Browser view of *fushi tarazu*.

**Figure supplement 16.** Browser view of *runt*.

**Figure supplement 17.** Browser view of *paired*.

**Figure supplement 18.** Browser view of *sloppy paired 1*.

**Figure supplement 19.** Browser view of *odd-paired*.

**Figure supplement 20.** Browser view of *huckebein*.

**Figure supplement 21.** Browser view of *tailless*.

**Figure supplement 22.** Browser view of *wingless*.

**Figure supplement 23.** Browser view of *engrailed*.

**Figure supplement 24.** Browser view of *patched*.

**Figure supplement 25.** Browser view of *hedgehog*.

**Figure supplement 26.** Browser view of *crocodile*.

468 of these enhancers (54.8%) are not transcriptionally active until hours later in development (*Figure 1C*). Notably, compared with peaks that open early (NC11), the set of peaks that open late (NC12-13) are enriched for enhancers that will be active later in development (>stage 4–6: n = 158 for NC11 and 310 for NC12 and 13: p=1.13x10$^{-7}$ by Fisher's Exact Test for ± NC11 by ± stage 4–6 expression).

In agreement with prior evaluations of the onset of zygotic gene expression (*Pritchard and Schubiger, 1996*; *De Renzis et al., 2007*; *Liang et al., 2008*; *Harrison et al., 2011*; *Lott et al., 2011*; *Chen et al., 2013*; *Ali-Murthy et al., 2013*; *Blythe and Wieschaus, 2015b*), we find that promoters for early expressed genes associated with the initial steps of embryonic patterning are in general open and accessible early, from NC11 onward. This includes the sets of gap genes (*giant, hunchback P1, Krüppel, and knirps,* see *Figure 1A* and *Figure 1—figure supplements 9–11*), pair-rule genes (*even-skipped, hairy, odd-skipped, fushi tarazu, runt, paired, sloppy paired 1,* and *odd paired*, see *Figure 1—figure supplements 12–19*), and terminal genes (*huckebein* and *tailless,* see *Figure 1— figure supplements 20* and *21*) that initially subdivide the embryonic anterior-posterior axis and termini. Likewise, we find that promoters for genes required to pattern the dorsal-ventral axis (*decapentaplegic, zerknullt, short gastrulation,* *brinker*, *snail*, and *twist,* data not shown) are scored open by NC11. Later acting embryonic patterning genes, such as the segment polarity genes, demonstrate variable timing for acquisition of chromatin accessibility, with the *wingless* promoter opening by NC11, but those of *engrailed* and *patched* open at NC12, and the *hedgehog* promoter becomes open by NC13 (*Figure 1—figure supplements 22–25*). Similarly, the 'head gap genes' also show variable acquisition of accessibility, with *orthodenticle* (*oc*), *cap'n'collar* and *buttonhead* opening by NC11 (data not shown), but *empty spiracles* opens at NC12, and *crocodile* opens at NC13 (*Figure 1—figure supplements 5* and *25*). A complete tabulation of recovered promoters and categorization of timing classes is provided in *Supplementary file 4*.

These two timing classes of chromatin regions are also differentially enriched for binding of Zelda and GAGA-factor (GAF), two transcription factors previously implicated in driving large-scale changes in chromatin structure during ZGA (*Blythe and Wieschaus, 2015b*; *Schulz et al., 2015*; *Sun et al., 2015*). Loci that are open early are enriched for binding of Zelda, whereas dynamic loci are enriched for binding of GAF (*Figure 1D*). These results suggest that the initial establishment of open chromatin proceeds in a stepwise or sequential manner where accessibility of genomic cis-regulatory elements precedes that of promoter and insulator elements. The coordinated change in promoter accessibility that occurs during NC12 and NC13 coincides with the large-scale recruitment of RNA Pol II to thousands of genes at ZGA (*Blythe and Wieschaus, 2015b*). These observations further support a model where the gain of promoter accessibility is the limiting factor for large-scale ZGA.

The timing of the MBT is controlled by at least two biological clocks, one which measures the relative nucleo-cytoplasmic (N:C) ratio and another which is N:C ratio independent and instead depends on elapsed developmental time (*Newport and Kirschner, 1982*; *Edgar et al., 1986*; *Edgar and Schubiger, 1986*; *Tadros et al., 2003*, *2007*; *Lu et al., 2009*; *Blythe and Wieschaus, 2015a*). We have previously predicted that chromatin structure is the primary target of regulation by these biological timers (*Blythe and Wieschaus, 2015b*; *Blythe and Wieschaus, 2015a*). If this were the case, then we would expect that delaying the onset of the MBT would lead to corresponding delays in acquisition of chromatin accessibility. To distinguish the relative contribution of each timer to establishment of chromatin accessibility, we performed an ATAC-seq timecourse on pairs of N:C ratio-matched haploid embryos maternally mutant for the *sesame* locus (*ssm*, flybase: *Hira*, see also Materials and methods) (*Loppin et al., 2000*; *Bonnefoy et al., 2007*). Haploid embryos undergo one additional pre-MBT mitotic division in order to achieve the same N:C ratio as their diploid counterparts (*Figure 2A*) (*Edgar et al., 1986*). Therefore, we predicted that any N:C ratio-dependent chromatin remodeling would be delayed in haploid compared with diploid embryos (*Lu et al., 2009*), whereas the timing of N:C ratio independent events would remain unchanged.

Open and accessible chromatin regions in haploids were identified on a per-cell-cycle basis as described above (n = 7217 total peaks in *ssm*). At equivalent developmental times, only 34.3 ± 8.1% of dynamic peaks are accessible in haploids compared with diploids. In contrast, at equivalent N:C ratios, 84.8 ± 11.8% of dynamic peaks are accessible in haploids versus diploids (*Figure 2B*, n = 4288 scored peaks). Based on this analysis, we classified each scored peak as N:C ratio- or time-dependent (n = 2630 and 1658, respectively). In diploid embryos, changes in accessibility occur with

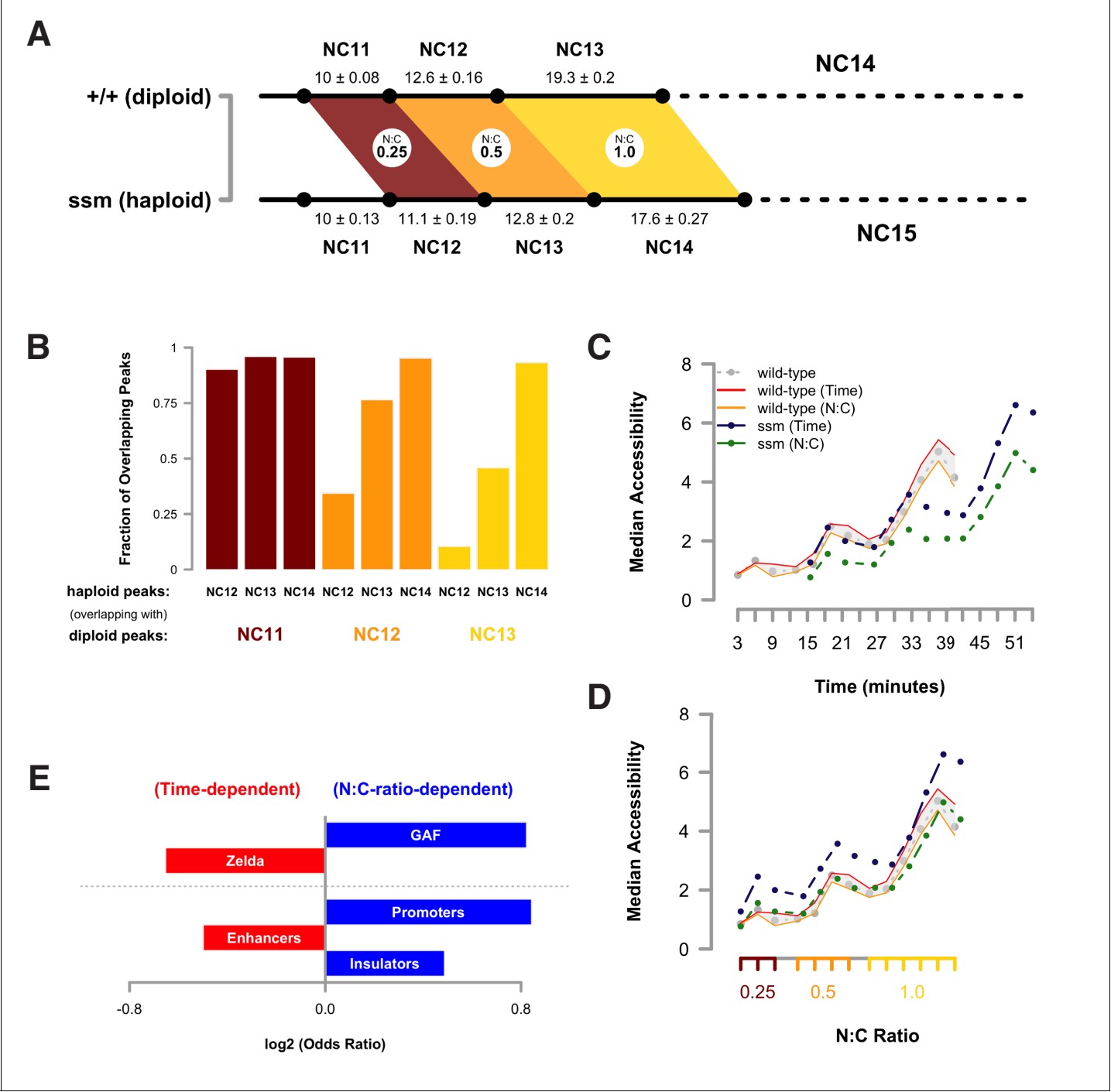

**Figure 2.** Dynamic acquisition of chromatin accessibility is regulated by the N:C ratio. (A) Cell cycle times for diploid (+/+, n = 21) and haploid embryos (*ssm*, n = 12) were measured by time lapse confocal imaging of His2Av-GFP. Mean cell cycle times ± SEM are indicated. Colored regions indicate periods of equivalent N:C ratios (0.25–1.0) between genotypes. (B) Accessible peaks were identified for haploid embryos and the fraction of overlap or co-occurence between diploids and haploids was calculated and plotted for the indicated cell cycles. (C) Median accessibility for haploid and diploid embryos was calculated for the sets of regions indicated in the legend (n = 2630 N:C ratio dependent regions, n = 1658 time-dependent regions). All differences in magnitude between time- and N:C ratio-dependent peaks in haploid embryos are statistically significant at p<0.01 by a randomization test, whereas only the NC13 + 15' time point achieves p<0.01 in the diploid dataset. Significance testing performed by proportionally dividing all datapoints into two groups at random and testing whether observed differences in median accessibility were equal to or greater than the difference observed between the 'N:C ratio' and 'time' groupings (one-tailed permutation test for n = 1×10⁴ trials). (D) Median accessibility data from panel C was re-scaled along the x-axis to match relative N:C ratios as shown in panel A. (E) Odds ratios for enrichment of the indicated genomic features and

*Figure 2 continued on next page*

*Figure 2 continued*

transcriptional regulators were calculated. Time-dependent chromatin accessibility is enriched for enhancers ($p=5.78 \times 10^{-7}$) and Zelda binding ($p=2.42 \times 10^{-12}$), whereas N:C ratio dependent chromatin accessibility is enriched for promoters ($p=5.66 \times 10^{-17}$), insulators ($p=2.42 \times 10^{-4}$), and GAF binding ($p=2.72 \times 10^{-09}$). p-Values are from two-sided Fisher's exact test on contingency tables constructed on [-/+ feature by time/N:C]. GAF odds ratios were calculated from regions that were GAF+ and Zelda-. GAF is significantly enriched in both N:C and time-dependent classes if co-occurrence with Zelda is considered.

The following figure supplements are available for figure 2:

**Figure supplement 1.** Examples of N:C-ratio-dependent regions.

**Figure supplement 2.** Examples of regions that gain accessibility independently of the N:C ratio.

similar dynamics in both time- and N:C-ratio-dependent loci. In haploids, time-dependent loci gain accessibility in advance of N:C-ratio-dependent loci (*Figure 2C*), which in turn only become accessible upon attaining the correct N:C ratio (*Figure 2D*). Importantly, classification into these different timing classes recapitulates the early versus late/dynamic dichotomy we observe between different genomic regulatory features (*Figure 2E*). Time-dependent loci are significantly enriched for enhancers and Zelda-associated regions (*Figure 2E*), similarly to the set of early accessible domains (*Figure 1D*). In contrast, N:C-ratio-dependent loci are significantly enriched for both promoter, insulator and GAF-associated regions (*Figure 2E*), similarly to the set of late or dynamic regions (*Figure 1D*). These results indicate that, similar to large-scale ZGA and cell cycle remodeling, the acquisition of chromatin accessibility at the MBT is likewise sensitive to the embryonic N:C ratio. Taken together, these results are consistent with a model where the biological timers that determine the MBT operate via distinct sets of chromatin regulatory factors such as Zelda and GAF in order to impart the sequential establishment of chromatin structure in advance of ZGA.

One general feature of chromatin we observe is that patterns of accessibility, once established, are maintained during both DNA replication and mitosis. We expected that large-scale chromatin condensation at metaphase would result in loss of accessibility during mitosis. This is not the case. Regulatory elements, such as promoters and enhancers (for example: *hb* P2 promoter/enhancer and shadow enhancer, *Figure 1A*, middle panel arrows) remain accessible even under conditions of large-scale chromatin condensation that accompanies mitotic metaphase (NC11 + 9', NC12 + 12', and NC13 + 18', see also *Figure 1—figure supplement 1*). Genome-wide, 73.5 ± 10% (mean ± sd) of all peaks maintain accessibility during metaphase (see also *Figure 1—figure supplement 2*). These results indicate that mitotic chromatin condensation does not necessarily erase prior accessibility states and that functional heritability of such states (*Ferraro et al., 2016*) across cell cycles could therefore stem from a simple mechanism for mitotic stability of chromatin accessibility.

When examined on a finer scale, established open regions display subtle cyclic periods of maximal and minimal accessibility that arise from DNA-replication-associated disruption and recovery of nucleosome structure. The period of minimal average chromatin accessibility correlates with the initial post-mitotic burst of DNA replication (*Blumenthal et al., 1974*), as estimated by the intensity of nuclear foci of PCNA-EGFP at replication factories (*McCleland et al., 2009*) (*Figure 3A*, see also *Figure 3—figure supplement 1* and Materials and methods). For example, during each cycle, the *hb* P2 promoter/enhancer displays minimum accessibility 3 min after entering interphase and subsequently reaches maximum accessibility during mitotic prophase (*Figure 3B*, red trace). The observed increase in overall accessibility during interphase positively correlates with the transcriptional activity of this locus, as measured by an MS2::MCP reporter (*Garcia et al., 2013*) (*Figure 3B*, blue trace). However, at mitotic metaphase, although transcriptional activity ceases (*Figure 3B*) and RNA Polymerase II is largely evicted from chromatin (*Figure 3C*), chromatin accessibility is nonetheless maintained (*Figure 3B*, grey bars). This suggests that the major constraint on the maintenance of chromatin architecture is nucleosome disruption during DNA replication and is consistent with a model where regulatory factors function to rapidly re-establish patterns of accessibility following passage of the replication fork.

Indeed, previous studies indicate that DNA replication can disturb patterns of nucleosome distribution, resulting in short-term interference with chromatin accessibility at newly replicated loci,

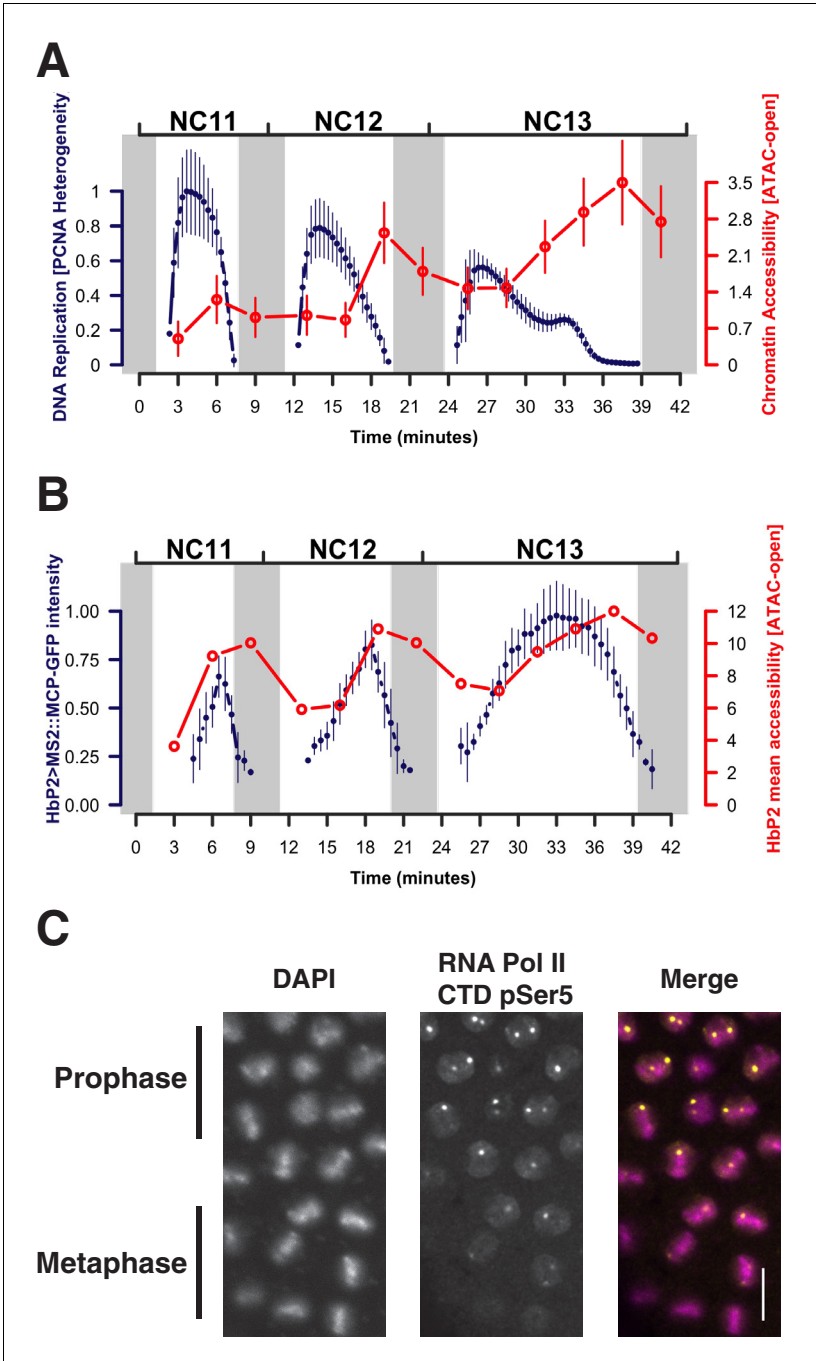

**Figure 3.** Patterns of stable and dynamic chromatin accessibility over the cell cycle. (**A**) Genome-wide mean ATAC-open coverage is plotted (red, error bars show std. dev. between peaks, n = 9824) over the mean scaled heterogeneity in PCNA-EGFP to estimate DNA replication activity (blue, n = 4, error bars show std. dev. between embryos). Mitotic phases are indicated by grey shading. (**B**) Mean coverage of sequencing reads from accessible chromatin over the *hunchback* P2 promoter/enhancer is plotted (red) over the mean measured spot fluorescence intensity from a *hunchback* P2> MS2(24) LacZ: : MCP-GFP reporter (blue, n = 3, error bars show std. dev. between embryos). Mitotic phases are indicated by grey shading. (**C**) Immunofluorecence for chromatin morphology (DAPI) and RNA Polymerase II (CTD pSer5) in a region of an NC13 embryo transitioning from prophase to metaphase is shown. Observed mitotic states are indicated at left. Scale bar = 10 μm.

The following figure supplement is available for figure 3:

**Figure supplement 1.** Example of PCNA-EGFP imaging and analysis.

which is counteracted by competition with regulatory factors that promote re-establishment of inter-phase chromatin architecture (*Ramachandran and Henikoff, 2016*). To measure how nucleosome positioning changes during these early embryonic cell cycles, we mapped nucleosome positions flanking the total set of accessible chromatin regions using NucleoATAC (*Schep et al., 2015*). One conserved feature of promoter chromatin architecture is a characteristic nucleosome-free region (NFR) upstream of the TSS. Promoters are useful for evaluating nucleosome positioning because, unlike in enhancers, the occupying nucleosomes adopt stereotypical phased positions around the NFR, anchored to a determined point of reference, the TSS, which facilitates the evaluation of dynamic nucleosome behavior at accessible sites following DNA replication.

In the Drosophila genome, on average, the +1 nucleosome is centered 135 bp downstream of the TSS, and the −1 nucleosome is centered 180 bp upstream of the TSS (*Mavrich et al., 2008*). Such NFR structures are observed at promoters between NC11 and NC13 (*Figure 4A* and *Figure 4—figure supplement 1B*). However, significant disruption to this promoter structure is clearly observed during DNA replication, particularly at 3 min into NC11 (*Figure 4A and B*, NC11 + 3′). Promoters largely recover nucleosome positioning by prophase and metaphase (*Figure 4B* and *Figure 4—figure supplement 1*, NC11 + 6′ and +9′, respectively). Nucleosome organization at promoters undergoes a similar disordering and recovery during NC12. However, the ability of replication to disrupt nucleosome structure is strikingly reduced during NC13, the developmental period when large-scale changes in zygotic transcriptional activity are first observed (*Figure 4A,B*, and *Figure 4—figure supplement 1*, NC13 + 3′ and +6′) (*Chen et al., 2013*; *Blythe and Wieschaus, 2015b*). This effect is also reflected in the times required for prior accessibility levels to be recovered at these regions following replication. On average, NFRs recover prior accessibility levels by 9 min into NC12. Despite the longer overall cell cycle time and incipient changes to the rate of S-phase progression during NC13, recovery of prior accessibility states is achieved on average by 6 min into NC13 (*Figure 4—figure supplement 2*). These results indicate that over this period there is a shift in the balance between replication-coupled nucleosome disordering and the counteracting regulatory factors that maintain patterns of chromatin accessibility. Initially, post-replicative gains in accessibility, although maintained through mitosis, are erased by DNA replication during the following cell cycle. Stabilization of nucleosome structure during NC13, however, would allow for large-scale recruitment of RNA Pol II to promoters early in the cell cycle. These observations are consistent with previous observations that the temporal regulation of the MBT stems from conflicting interactions between RNA transcription and DNA replication, at the level of the initial establishment of accessible chromatin structure (*Blythe and Wieschaus, 2015b*).

We next addressed whether the resistance to replication-coupled nucleosome disruption acquired by open regions at NC13 arises independently of the changes DNA replication rate and origin firing that also occur at this cycle (*Shermoen et al., 2010*; *Farrell and O'Farrell, 2014*). To distinguish these possibilities, we measured changes in NFR size between time- and N:C-ratio-dependent promoters in haploid embryos (*Figure 4B*). The average NFR size for N:C-ratio-dependent promoters in haploids correlates well with that of diploid embryos when plotted over equivalent N:C ratios (*Figure 4B*, blue trace), demonstrating attenuated replication coupled nucleosome disruption during the final cell cycle before the MBT. In contrast, the set of time-dependent promoters in haploids demonstrate attenuated disruption of NFR size one cell cycle earlier than their N:C ratio dependent counterparts, prior to S-phase remodeling (*Figure 4B*, red trace, asterisks). Likewise, in haploid embryos, early, time-dependent Zelda-associated promoters demonstrate increased resistance to replication-associated nucleosome disruption compared with later, N:C-ratio-dependent GAF-associated promoters (*Figure 4C*). This indicates that resistance to replication-coupled nucleosome disruption occurs independently of S-phase remodeling and is linked to the mechanisms that drive the initial establishment of accessible states.

Taken together, our observations are consistent with a model for ZGA where chromatin accessibility at genomic features is conferred in defined steps and that establishment of particular states depends on competition between DNA replication machinery and cis-regulatory factors. Such competition is not limited to embryonic tissues but is likely a general mechanism for recovery of accessibility states following replication (*Ramachandran and Henikoff, 2016*). Our results extend these observations by indicating that such competition may be a major point of regulation for the developmental acquisition of stable patterns of chromatin accessibility. Our results also indicate that stability of nucleosome positioning during mitosis is a major mechanism for epigenetic inheritance of prior

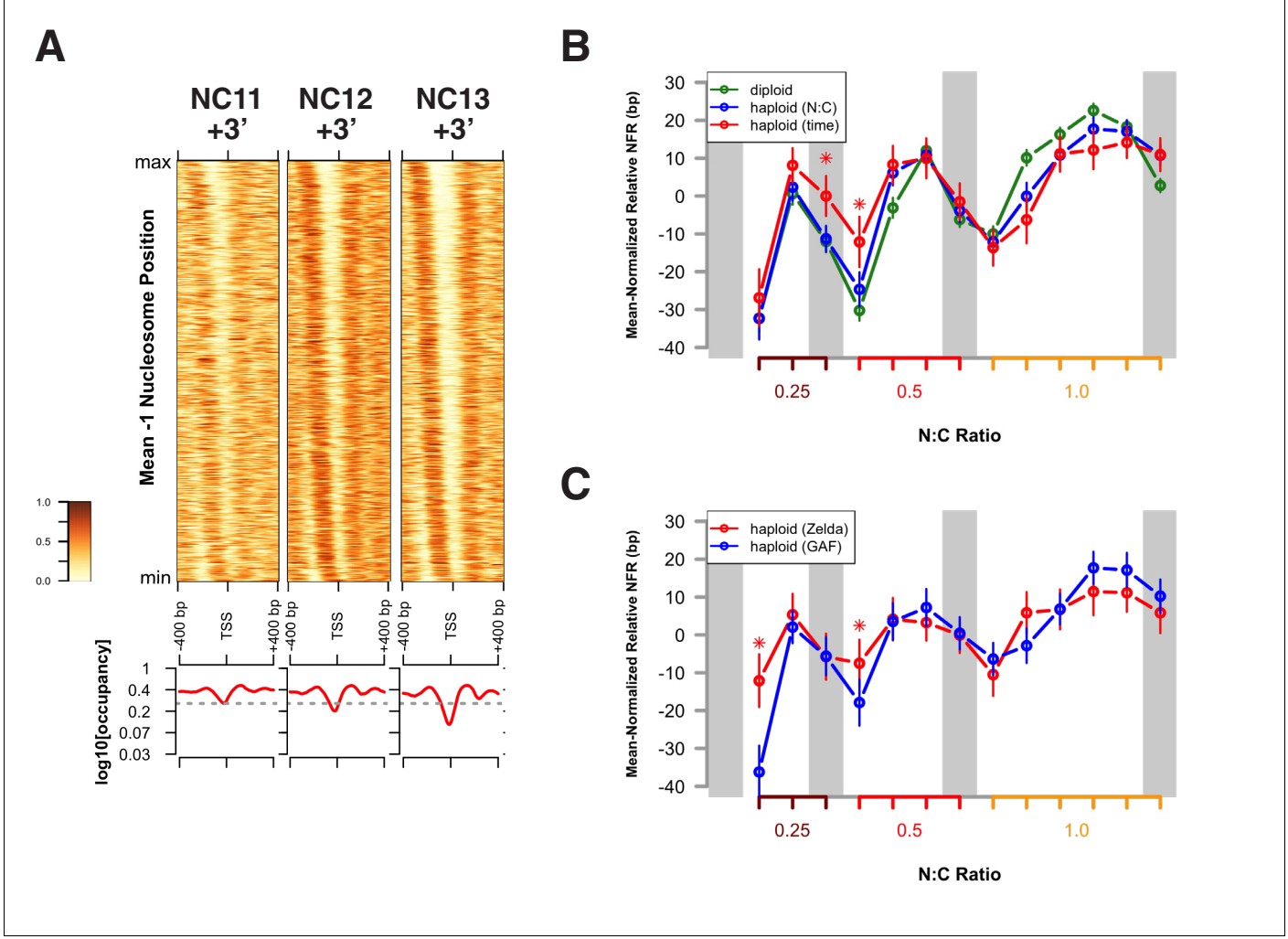

**Figure 4.** Attenuation of replication coupled nucleosome disruption at the MBT. (**A**) Predicted nucleosome occupancy is plotted for the selected time points (top) for the 800 bp region flanking a set of early embryonic promoters, centered over the TSS. These timepoints correspond to the initial phases of DNA replication in each cell cycle. Nucleosome profiles for the complete timecourse are provided in *Figure 4—figure supplement 1*. Promoters are ordered on the y-axis by the average −1 nucleosome position over the entire time course, and the relative nucleosome occupancy is represented by the colorbar at left. The log10 average occupancy signal for each time point is plotted below each heatmap (red). Maximum nucleosome signal at the TSS over the entire timecourse is indicated by the grey line. (**B**) Mean-normalized relative NFR sizes for time (red) and N:C ratio (blue) dependent loci are plotted for haploid embryos for comparison with diploid embryos (green). Data are plotted as a function of N:C ratio (x-axis). NFRs for time-dependent loci demonstrate minimal closing during metaphase (N:C = 0.25, p<0.01, asterisk) and early S-phase (N:C = 0.5, p<0.01, asterisk) compared with N:C-ratio-dependent loci and N:C ratio matched diploid loci. p-Values indicate frequencies of observing differences between randomly selected loci greater than or equal to that observed for the time- and N:C-ratio-dependent groups (one-tailed permutation test for n = 1×10⁴ trials). Error bars show the 95% confidence interval for differences between the plotted median values. (**C**) Mean-normalized relative NFR sizes for Zelda-associated (red) and GAF-associated (blue) promoters was plotted for haploid embryos. Plotting and significance testing is as described for panel B. Zelda-associated promoters demonstrate increased NFR stability during early S-phase during both N:C = 0.25 and N:C = 0.5 (asterisks, p<0.01, one-tailed permutation test for n = 1×10⁴ trials). Error bars show the 95% confidence interval for differences between the plotted median values.

The following figure supplements are available for figure 4:

**Figure supplement 1.** Nucleosome positioning over promoters.

**Figure supplement 2.** Changes in accessibility at nucleosome-free regions over the cell cycle.

chromatin states from one cell cycle to the next. Whether such stability requires an active mechanism for stabilization, as has been proposed in mammalian systems (*Blobel et al., 2009*; *Hsiung et al., 2015*), remains to be determined. Our model therefore predicts that the initial establishment of stable accessible chromatin states during periods of intense mitotic activity is limited by the availability of transcription factors. Upon reaching a critical activity threshold, regulatory factors such as Zelda and GAF would successfully compete with nucleosomes to efficiently re-establish patterns of chromatin accessibility following DNA replication. This competition model is consistent with a prior observation that reduction of function for either GAF or Zelda is sufficient to reduce conflicts between the DNA replication machinery and some feature of chromatin remodeling at ZGA (*Blythe and Wieschaus, 2015b*). Such competitive interactions between systems for transcription and replication may also be conserved timing mechanisms in vertebrate embryos (*Almouzni and Wolffe, 1995*; *Amodeo et al., 2015*).

## Materials and methods

### Model organism

All reported experiments were performed on embryos from transgenic and mutant variants of the Fruit Fly *Drosophila melanogaster* (NCBI Taxon 7227).

### ATAC-seq sample preparation

Embryos produced from either wild-type (*w; His2Av-GFP*) or homozygous *sesame* (*w ssm^185b; His2Av-GFP*) mothers were collected on yeasted apple juice agar plates. Individual embryos were selected from plates and arrayed in individual wells in Nunc Microwell Mini Trays (VWR, Radnor, PA), and nuclear morphology was observed by Histone-GFP fluorescence under an AMG EVOS FL digital microscope at 20x magnification. Temperatures for sample collection were maintained between 21°C and 24°C to minimize variation in cell cycle timing (*Supplementary file 2*). Under these conditions, cell cycle times were indistinguishable from those measured by confocal microscopy and reported in *Figure 4A* (*Blythe and Wieschaus, 2015b*). Developmental stage was determined as follows: NC12 = 12 to 18 nuclei per 2500 $\mu m^2$; NC13 = 22–30 nuclei per 2500 $\mu m^2$; and NC14 = > 30 nuclei per 2500 $\mu m^2$. Cell cycles were timed from the onset of anaphase of the previous cell cycle. The staging of samples at 3-min intervals relies on the synchrony of nuclear divisions and the extraordinary reproducibility in the timing and duration of cell cycles in the cleavage stage Drosophila embryos. Each embryo was hand-selected for analysis at metaphase of the preceding cycle. To ensure synchrony, only embryos in which all nuclei had entered metaphase within a 30 s window were selected. Anaphase was then required to have begun across the length of the embryo within a 60 s window and the total elapsed time for mitosis (metaphase start to anaphase start) was not allowed to exceed 3 min. Embryos exceeding these limits ( ~10%) were discarded. Under these conditions, embryos selected at cycle 10 anaphase reach metaphase of cycle 11 in 8.2 ± 0.4 min ± s.d., embryos selected at cycle 11 anaphase reach metaphase of cycle 12 in 10.7 ± 0.9 min ± s.d., embryos selected at cycle 12 anaphase reach metaphase 13 in 17.2 ± 0.9 min ± s.d. This precision allows us to select and process embryos for each metaphase stage by selecting at the previous anaphase. Consistent with the reproducible progression of the Drosophila cell cycles, embryos processed at metaphase and timed from the previous cycle interval cluster with each other and single metaphase embryos selected in the previous cell cycle show highly similar and correlated patterns of accessibility (see *Figure 1—figure supplement 1*).

Upon selecting an embryo, a timer was started upon observation of anaphase of the cell cycle prior to the target collection stage. Because processing embryos for ATAC-seq requires approximately 2.5 min, to collect at NC12 + 6 min, for example, a previously selected embryo was collected from its well at 3.5 min post-anaphase. It was then dechorionated by a 45 s exposure to freshly prepared 4% bleach (Clorox) followed by 3 x ~6 ml washes in deionized water. Subsequently, the embryo was placed inside a detached cap of a 1.5-ml microcentrifuge tube and at 6 min post-anaphase, the embryo was overlaid with 10 μl of ice cold Lysis Buffer (*Buenrostro et al., 2015*) (10 mM Tris-HCl pH 7.4; 10 mM NaCl; 3 mM MgCl$_2$; 0.1% Igepal CA-630) and immediately macerated to homogeneity with a fire-polished microcapillary tube (Drummond Microcap 20 μl, Sigma-Aldrich, St. Louis, MO). The cap was next inverted and placed over a 1.5 ml low-retention

microcentrifuge tube (Eppendorf) containing an additional 40 µl of Lysis Buffer and spun for 10 min at 500 RCF at 4°C. Supernatant removal was monitored under a stereomicroscope to ensure that the crude nuclear pellet, visible as a yellow-grey mass, was not disturbed. The nuclear pellet was then placed in dry ice prior to further processing until several additional samples were collected. All samples were frozen for at least 30 min prior to fragmentation and library preparation. In a set of optimization experiments (data not shown), freezing the pellet for short periods of time had no effect on the overall efficiency or quality of ATAC-seq data.

Fragmentation and amplification of ATAC-seq libraries were performed essentially as described previously (*Buenrostro et al., 2015*). Fragmentation reaction volumes were 10 µl for a single embryo, and used 2.5 µl of Tn5 Transposase from the Illumina Nextera Sample Preparation Kit. The volume of Tn5 was determined empirically, where 0.5 µl produced insufficient fragmentation, and 5 µl was over-fragmented. Fragmentation reactions took place for 30 min on an Eppendorf Thermomixer set to 37°C and 800 rpm. Samples were purified through Qiagen Minelute columns following the manufacturer's protocol for enzymatic reaction clean up, eluting in 10 µl. Optimization experiments indicated that this purification was essential for high-quality ATAC-seq libraries from single embryos, although this step is considered 'optional' in published protocols (*Buenrostro et al., 2015*).

Library amplification was performed as described previously (*Buenrostro et al., 2015*). All libraries amplified with a range of 11–14 total PCR cycles, and the number of cycles roughly corresponded to developmental stage, where older embryos required fewer and younger embryos required more cycles (data not shown, see *Supplementary file 2*). Amplified libraries were purified from PCR reactions with 1.8x Ampure SPRI beads following the manufacturer's instructions and 1 µl of a 1:10 diluted library prep was evaluated on a Bioanalyzer HS-DNA chip to estimate quality. Concentrations were estimated using a Qubit fluorometer. To control for sample loss or contamination during preparation, samples showing significant deviations were discarded, based on final library concentration, known DNA content of the starting material, total number of PCR cycles, and the overall variance of these values for all libraries for a single experiment.

For wild-type NC12 and NC13 samples, single embryos were collected for each time point (*Supplementary file 1*). The wild-type NC11 sample set was collected subsequently, with the purpose of comparing directly with the wild-type NC12 data set, where we observe a transitional state of chromatin structure. To facilitate this direct comparison, two NC11 embryos were collected per time point, per replicate, in order to minimize differences introduced by reduced genomic nuclear content. Similarly, the haploid NC12 and NC13 samples were collected with the intent of direct comparison with wild-type NC12 and NC13. As such, two haploid embryos were collected per time point, per replicate. Single haploid NC14 embryos were collected per time point per replicate. As such, samples ranged between 4096 and 8192 estimated diploid genomic equivalents (*Supplementary file 2*).

For samples where two embryos were collected per time point, per replicate, individual embryos were collected and processed through the fragmentation step described above. Samples were combined during the Qiagen Minelute clean up step. This approach enabled samples to be fragmented under identical conditions. During optimization experiments, combination of two embryos in a single 10 µl fragmentation volume resulted in inefficient fragmentation compared with individual embryos, and could stem from increased non-nuclear/genomic contaminating material per unit volume of fragmentation reaction.

We note that the input sample to the fragmentation reaction contains not only genomic DNA, but also a significant proportion of mitochondrial DNA. During library preparation, we made no effort to separate nuclear and mitochondrial fractions. During early development (0–2.5 hr), mitochondrial DNA comprises between 50 and 99% of the total DNA content in the egg. Replication of mitochondrial DNA is not coupled to that of nuclear DNA, and whereas nuclear DNA amplifies exponentially in early development, mitochondrial DNA does not replicate until much later (>6 hr) (*Rubenstein et al., 1977*). Therefore, since mitochondrial DNA content remains effectively constant during the period of sample preparation and is abundantly recovered, this conveniently 'buffers' the library preparation despite the large changes in genomic, nuclear DNA content.

Attention was given to sample randomization, where different timepoints were collected on different days whenever possible. Likewise, distribution of libraries within pools was designed to

minimize inclusion of identical timepoints and identical library prep dates within the same pool. Sample collection, library preparation, and analysis was not blinded.

When sequenced to extreme depth, ATAC-seq can reveal 'footprints' of transcription factor binding sites (*Buenrostro et al., 2013*). Although we did not formally attempt to measure whether transcription factor footprints are resolved in this data set, it is likely that these sequencing libraries are not of sufficient depth to reliably detect this phenomenon. On average, our libraries were sequenced to a depth ($\sim 10^7$ reads) that was sufficient for the purposes of calling peaks and distinguishing nucleosome positions. However, this depth is an order of magnitude less than the recommended depth for resolving footprints ($\sim 10^8$ reads) (*Buenrostro et al., 2015*).

Pooled, barcoded ATAC-seq libraries were subjected to 2×67 bp paired end sequencing on an Illumina HiSeq 2500 at the Lewis-Sigler Institute for Integrative Genomics Sequencing Core Facility, Princeton, NJ.

## Data pre-processing

Barcode-split sequencing files were first subjected to adapter trimming, using TrimGalore version 0.4.0 (www.bioinformatics.babraham.ac.uk) with the optional parameters –trim1 and –paired.

Trimmed reads were mapped to the Drosophila melanogaster genome (dm6) using BWA (*Li and Durbin, 2009*) (http://bio-bwa.sourceforge.net). First, individual read ends were mapped using BWA aln with default parameters. Paired-end mapped reads were then compiled using BWA sampe with option –a 5000.

Mapped reads were further processed using samtools (*Li et al., 2009*) (http://www.htslib.org). BWA output was imported using samtools import with default parameters. Imported reads were sorted (samtools sort), and indexed (samtools index). Reads likely originating from PCR or optical duplicates were marked using Picard MarkDuplicates (https://broadinstitute.github.io/picard/) using default parameters.

Reads were filtered using samtools view. Reads were required to have a map quality score greater than or equal to 30 (-q 30), to be mapped, to be proper pairs, and not to be secondary mappings. Duplicates were also removed.

Paired-end reads were subsequently imported into R as a GenomicRanges (*Lawrence et al., 2013*) (https://bioconductor.org) object. Only reads mapping to chrX, chr2L, chr2R, chr3L, chr3R, and chr4 were used for downstream analysis, although chrY was also imported to sex single-embryo samples. Fragment ends were adjusted to reflect the original Tn5 interaction site, as described previously (*Buenrostro et al., 2013*): Watson strand start sites had four base pairs subtracted, and Crick-strand start sites had five base pairs subtracted. In practice, this is executed on a GRanges object where paired end reads have been collapsed into single intervals by calling: start(granges.object) - 4; end(granges.object) + 5.

Biological replicates for individual time points were merged for subsequent analysis.

To distinguish reads originating from open/accessible chromatin reads, samples were fit to a mixed exponential/Gaussian distribution as described previously(*Buenrostro et al., 2013*). Based on these fits, a ≤98 bp cutoff was selected to identify reads originating from open chromatin regions. Merged timepoint datasets were filtered for reads of width ≤98 bp and designated as 'open' reads.

Such reads were either kept in GenomicRanges format for further analysis, or were exported as bed files for peak-calling by Zinba (*Rashid et al., 2011*).

## Read length distribution and estimation of stage-specific library preparation bias

To examine the effect of cell cycle stage on the recovery of sequenced fragments, we plotted histograms of the distribution of recovered fragment sizes (*Figure 1—figure supplement 3*). As discussed in the introduction, the estimate of accessibility and nucleosome positioning by ATAC seq is sensitive to chromatin topology, where these features can be measured reliably in generally 'open' chromatin, but that the expected reduced recovery of small accessible fragments in generally 'closed' chromatin could limit the resolution of the data. Given that mitotic chromatin, in particular, is superficially expected to have a compact and largely inaccessible structure, we paid close attention to the relative recovery of small and large sequencing fragments for these stages. We observe that indeed, a greater proportion of large, nucleosome-protected fragments are recovered in both

metaphase and early interphase (+3 min) samples (*Figure 1—figure supplement 3*). The reason for this distribution could be biological in nature: a greater proportion of the genome may be stably associated with nucleosomes during metaphase and early interphase compared with later times. However, we were careful to consider that this distribution could instead reflect an artifact. In principle, during periods of intense chromatin compaction, a significantly smaller proportion of genomic DNA could perhaps be fragmented by Tn5, but since libraries are PCR-amplified to similar final concentrations, then relatively fewer unique 'open' regions would be recovered than expected. We predicted that if such an artifact were present, that we would have less fragmented DNA in samples originating from metaphase stages. However, as discussed above, the abundance of mitochondrial DNA effectively buffers samples from wide variation in input DNA content. In this case, we would nonetheless expect to observe fewer overall unique recovered sequence reads that map to genomic DNA, if a significant artifact was present in the metaphase samples. To address this possibility, we plotted the inferred quantity of initial fragmented DNA from our individual library preps, and the number of unique sequence reads recovered for each library (*Figure 1—figure supplement 3*). Although there is a trend for metaphase samples on average to have less starting fragmented DNA content and fewer uniquely mapping reads, the magnitude of the difference is small, and indistinguishable from a random permutation of samples. We were therefore confident that open regions recovered from metaphase samples do in fact reflect regions of chromatin accessibility and are minimally influenced from major factors pertaining to sample preparation.

## Peak calling

Sample sizes (n $\geq$ 3 biological replicates per timepoint) were sufficient to yield on average 15 million unique sequencing reads per pooled timepoint that map to the canonical chromosome assemblies (dm6) (see *Supplementary file 1*). We found that peak-calling was sensitive to sequencing read depth (data not shown), where reproducibility was significantly affected when fewer than 17.5 million reads were used. Therefore, we pooled samples by cell cycle, or by genotype (all timepoints) in order to call peaks on open chromatin reads.

Peaks were called using Zinba (*Rashid et al., 2011*) version 2.03.1 (https://code.google.com/archive/p/zinba/issues/69). To call peaks for any individual cell cycle, merged open datasets for all timepoints within an individual cell cycle were combined. Likewise, all open chromatin datasets were merged together for calling peaks on the entire dataset for a particular genotype.

A custom mappability file for *Drosophila melanogaster* genome assembly dm6 with 67 bp reads was generated using the Gerstein Mappability Map software (*Rozowsky et al., 2009*) (http://archive.gersteinlab.org/proj/PeakSeq/Mappability_Map/Code/). For paired-end reads, extension values for generateAlignability(), basealigncount(), and run.zinba() were set as the average fragment size in the set of reads originating from input open chromatin datasets, 65 $\pm$ 2 bp.

Peaks were called using run.zinba() with the following optional parameters: input='none', winSize=300, offset=50, extension=(see above), selectmodel=FALSE, formula=exp_count~exp_cnvwin_log +align_perc, formulaE=exp_count~exp_cnvwin_log+align_perc, formulaZ=exp_count~align_perc, FDR=TRUE, threshold=0.05, winGap=0, cnvWinSize=7.5E + 4, refinepeaks=TRUE.

Parameters were selected to optimize resolution of closely positioned regions of open chromatin, e.g. the *even-skipped* locus on chr2R. With these parameters, individual closely-packed known enhancers, insulators, and coding elements are effectively distinguished.

Following peak calling, only regions with posterior probability scores $\geq$0.8 were retained.

To estimate sample reproducibility within biological replicates from identical timepoints (*Figure 1—figure supplement 1*, *Supplementary file 1*), CPM normalized counts of open chromatin fragments within peaks were summed and correlation coefficients and standard deviations between replicates were calculated.

## Open chromatin coverage calculation and normalization

Example R code for calculating coverage and performing normalization are provided as a supplementary file.

To calculate coverage of open chromatin reads, the average number of 'open' reads within 10 bp windows tiling the chromosomes of interest was calculated. Coverage values were normalized to the counts per million of the original, unsplit ('open' and >98 bp read) dataset. The rationale for

normalizing to the full dataset rather than to just open reads was that this would likely minimize artificial inflation of open read coverage in cases where open reads comprised a smaller fraction of the overall dataset.

Finally, all open chromatin coverage measurements were normalized by standardization to the mean and standard deviation of coverage over a set of 25,000 randomly selected background regions. To select background regions, the set of peak open regions were widened to 20,000 bp, reduced, and subtracted from the genome assembly. Thereafter, 25,000 random positions were selected and widened to reflect the distribution of widths in the set of open peaks. Coverage within these background regions was then calculated, and regions with zero coverage were discarded (~5%). The distribution of counts within background regions approximated a log-normal distribution. Mean and standard deviation of these background regions was calculated and used to transform the coverage measurements for the entire genome.

This normalization strategy relies on the assumption that randomly selected background regions would have a similar likelihood of background fragmentation during sample preparation, regardless of cell cycle stage or developmental stage. In practice, this transformation has the beneficial effect of setting a consistent floor for the data across all timepoints. During this analysis, all major conclusions derived from the open chromatin datasets were confirmed with both simple CPM normalization, and this background standardization and were determined to be essentially equivalent.

## Assigning open peaks to genomic features

Peaks were assigned to either promoter, insulator, or enhancer categories by identifying peaks that overlap known genomic annotations (*Harrison et al., 2011*; *Celniker et al., 2009*; *Kvon et al., 2014*). A peak was assigned to the 'promoter' category if any or all of the peak overlapped with the 50 bp region flanking the transcriptional start site. Peaks were assigned to the 'insulator' category if a peak was not assigned to the 'promoter' class, and if any or all of a peak overlapped with peaks bound by two or more of the following insulator proteins: CTCF, GAF, BEAF32, CP190, Mod(Mdg4), and Su(Hw). Finally, peaks were assigned to the 'enhancer' category if a peak was not assigned to the 'promoter' or 'insulator' class, and if any or all of a peak overlapped with peaks associated with either a published set of experimentally validated enhancers (*Kvon et al., 2014*) or two or more of the following chromatin marks/ transcription factors: CBP, Histone H3K4me1, Histone H3K27ac, or Zelda. The rationale for selecting this panel of factors to categorize putative enhancers was that the set of functionally validated enhancers is not complete and represents less than 20% of the genome. The CBP/K4me1/K27ac set of marks is typically used to identify active enhancers in other genomics studies (*Heintzman et al., 2009*; *Creyghton et al., 2010*; *Zentner et al., 2011*; *Rada-Iglesias et al., 2012*; *Calo and Wysocka, 2013*). Finally, Zelda binding outside of promoter regions is often associated with highly occupied target regions that often attract additional transcription factors and these regions are typically over-represented for developmentally active enhancer regions [(*Kvon et al., 2012*) and references therein)]. Peaks that did not fall into one of these three categories were categorized as 'other'.

Datasets from modEncode were downloaded from (http://data.modencode.org) as peak lists. Genomic coordinates were converted to the dm6 assembly and combined: CTCF 0–12 hr ChIP-chip (769 and 770); GAF 0–12 hr ChIP-chip (23); BEAF32 0–12 hr ChIP-chip (21); CP190 0–12 hr ChIP-chip (22); Mod(Mdg4) 0–12 hr ChIP-chip (24); Su(Hw) 0–12 hr ChIP-chip (27 and 901); Nej/CBP 0–4 hr ChIP-chip (875 and 900); H3K4me1 0–4 hr ChIP-chip/seq (423 and 777); H3K27ac 0–4 hr ChIP-chip/seq (424, 970 and 834).

## Haploid embryos

Embryos produced from mothers mutant for *sesame/Hira* (*ssm[185b]*) are fertilized normally but are unable to de-condense the male pronucleus during the first zygotic cell cycle (*Loppin et al., 2000*). These embryos subsequently develop as haploids and therefore have altered N:C ratios at equivalent elapsed times during early development. *sesame* mutant embryos are morphologically normal during the early stages of development through gastrulation, and a significant proportion of embryos progress to form cuticle. We chose to generate haploid embryos using *sesame*, because the two other mutants that yield haploid embryos (*maternal haploid*, and *ms(3)K81*) develop with additional, uncharacterized defects that affect overall cell cycle duration, and high rates of DNA

damage and abnormal nucleokinesis ([*Edgar et al., 1986*], and data not shown). We expected that these additional phenotypes would confound the temporal comparison between diploid and haploid embryos. *sesame* mutants, in contrast, develop with nearly indistinguishable cell cycle rates and overall fidelity of nuclear division compared with wild-type embryos (*Figure 2A*, and data not shown). The product of the *sesame* locus, Hira, is a histone chaperone responsible for the DNA replication independent deposition of the histone variant H3.3 (*Ray-Gallet et al., 2002*). H3.3 is typically deposited in regions of open, transcriptionally active chromatin. Notably, the primary essential function of Hira in early embryos is to facilitate male pronuclear decondensation: H3.3 deposition is grossly unaffected in *sesame* mutant embryos, indicating a Hira-independent mechanism for H3.3 deposition in early embryos (*Bonnefoy et al., 2007*). On the basis of these observations, we expected that differences observed between wild-type and *sesame* mutant embryos would primarily reflect alterations in embryonic DNA content, and would be minimally confounded by H3.3-dependent effects on chromatin accessibility. At present, however, we are unable to rule out any H3.3-independent effects of Hira on the acquisition of chromatin accessibility in our experimental results.

## Classification of N:C-ratio-dependent and –independent peaks

Peaks were called on *ssm* datasets as described for wild-type. For each cell cycle, we calculated the co-occurrence of peaks in either *ssm* or wild type. To assign a peak to either the N:C-ratio-dependent or –independent class, wild-type peaks that gain accessibility in either NC12 or NC13 (late or dynamic peaks) only were scored, and a peak was required also to be 'present' in the set of *ssm* peaks. We consistently observed that 27% (2654 of 9824) of accessible peaks in wild type are not called open in *ssm* mutant embryos at similar degrees of sequencing coverage. These 'missing' peaks were not assigned to any timing category (see *Supplementary file 3*), nor were wild type peaks that were designated 'Open by NC11'.

## Nucleosome positioning

Nucleosome positions were calculated using the NucleoATAC software package (*Schep et al., 2015*) (https://github.com/GreenleafLab/NucleoATAC) version 0.3.1 using default parameters. Zinba peak regions were widened to 2500 bp, centered over the maximum read density value calculated, and used as input regions of interest for NucleoATAC analysis. BAM-formatted reads were used, wherein duplicate-filtered reads mapping to canonical chromosomes from biological replicates were pooled by timepoint and genotype.

NucleoATAC returns the predicted positions and occupancy scores of nucleosomes based on the coverage of nucleosome-protected ATAC fragments delimited by flanking 'open' ATAC fragments. Following nucleosome position calling, predicted dyad centers were modeled into nucleosomes by widening to 147 bp. NFRs were calculated by finding the predicted NucleoATAC dyad centers immediately upstream (−1) and downstream ( +1) of the position 25 bp upstream of the TSS. We found that using the −25 position for calculation of the −1 and +1 nucleosomes was necessary because the +1 nucleosome can at certain timepoints occlude the TSS, and in these circumstances using the TSS as the vantage point for NFR identification will mistakenly identify the +1 and+2 nucleosome. For certain highly expressed genes (e.g., *slam, bnk, sry-alpha*) a +1 nucleosome was not called by NucleoATAC. For approximately 33% of all promoters, either the −1 or +1 nucleosome was not called by NucleoATAC. In these cases, the promoter was omitted from further analysis. Plots in *Figure 4A* show the distribution of modeled nucleosomes weighted by the calculated occupancy score reported by NucleoATAC. Plots in *Figure 4B* and C show the relative changes and deviations in NFR size between different genotypes and subgroupings, the NFR measurements were mean-normalized such that the average NFR size per group has a 0 bp size. Non-mean normalized NFR sizes (i.e., raw average base-pair distances) for wild-type are presented in *Figure 4—figure supplement 1*.

## EGFP-PCNA imaging and analysis

EGFP-PCNA was produced by via a transgene consisting of a genomic fragment sufficient to complement PCNA loss of function (dm6 chr2R:20261602–20263418) into which an N-terminal Drosophila codon optimized EGFP cassette and linker sequence was engineered (sequence available upon request). Data were collected by time-lapse laser scanning confocal microscopy of embryos

expressing Histone H2Av-RFP and EGFP-PCNA at a 15 s frame rate with a 63×1.4 NA oil immersion objective. Z-stacks were collected to cover the entire span of cortically localized nuclei between NC10 and NC14 at 0.5 µm intervals. Images were maximum-projected for analysis with Matlab (2014A). PCNA tracks along with DNA Polymerase during replication and the GFP-fusions appear as bright, heterogeneously-distributed foci within S-phase nuclei (*Figure 3—figure supplement 1*) (*McCleland et al., 2009*). To estimate the overall timing of S-phase between NC11 and NC13, we quantified the degree of PCNA heterogeneity over time. Nuclei were segmented on the Histone RFP channel, and heterogeneity in EGFP-PCNA signal was measured using a grayscale correlation matrix (*Figure 3—figure supplement 1*). EGFP-PCNA signal was down-sampled to six grayscale levels and correlation matrices were calculated for radial sets of eight pixels separated by 200 nm distance, using the Matlab function graycomatrix.m. For each frame, heterogeneity was calculated as 1 – 'Homogeneity', as reported by graycomatrix.m. This yields a value between 0 and 1 where 0 indicates that all pixels neighbor pixels of similar intensities, and one indicates that all pixels neighbor pixels of different intensities. The by-frame heterogeneity measurements were next scaled to the mean EGFP-PCNA fluorescence intensity to yield an estimate of the fraction of PCNA associated with heterogeneously distributed fluorescence intensity. We note that similar results have been reported using a different quantification approach that measures the overall coefficient of variation in nuclear PCNA fluorescence intensity rather than grayscale correlation (*Deneke et al., 2016*).

### MS2::MCP-GFP reporter imaging and analysis

Embryos from a cross between *w; Histone H2Av-RFP/+; MCP-GFP(4F)/+* females and *y w; HbP2-promoter>MS2(24)-LacZ* males (*Garcia et al., 2013*) were imaged by laser scanning confocal microscopy at a 30 s frame rate with a 63×1.4 NA oil-immersion objective from NC10 through NC14. Z-stacks were collected at 0.5 µm intervals for a 12 µm cortical region of the embryo. Image processing and analysis was performed as described previously (*Garcia et al., 2013*).

### Cell cycle timing

Cell cycle timing for *w ssm185b; His2Av-GFP* was performed as described previously. Wild-type cell cycle times reflect data reported previously (*Blythe and Wieschaus, 2015b*).

### Data availability

Raw sequence files, coverage files, nucleosome positions, and an annotated peak list have been deposited in Gene Expression Omnibus GSE83851.

## Acknowledgements

We thank S Di Talia, P Klein, M Levo, and S Little for comments on the manuscript; A Amodeo, K Chen, T Gregor, C Hannon, M Levine, P Schedl, and T Schüpbach for helpful discussions; H Garcia for assistance with quantification of MS2-MCP data; and W Wang and the staff of the Lewis-Sigler Institute Sequencing Core Facility. This work was supported in part by NIH grant number R37HD15587 (EFW) and Ruth Kirschstein NRSA Postdoctoral Fellowship F32HD072653 (SAB). EFW is an investigator with the Howard Hughes Medical Institute.

## Additional information

### Funding

| Funder | Grant reference number | Author |
|---|---|---|
| National Institutes of Health | F32HD072653 | Shelby A Blythe |
| Howard Hughes Medical Institute | | Shelby A Blythe<br>Eric F Wieschaus |
| National Institutes of Health | R37HD15587 | Eric F Wieschaus |

The funders had no role in study design, data collection and interpretation, or the decision to submit the work for publication.

### Author contributions
SAB, Conception and design, Acquisition of data, Analysis and interpretation of data, Drafting or revising the article; EFW, Analysis and interpretation of data, Drafting or revising the article

### Author ORCIDs
Eric F Wieschaus, http://orcid.org/0000-0002-0727-3349

# Additional files

### Supplementary files
• Supplementary file 1. Sample summary.

• Supplementary file 2. Sample metadata.

• Supplementary file 3. Annotated peak regions used in the analysis.

• Supplementary file 4. Annotated promoter regions.

• Supplementary file 5. Code example for data normalization in R markdown (.rmd) format.

• Supplementary file 6. Code example for data normalization in. pdf format.

### Major datasets
The following dataset was generated:

| Author(s) | Year | Dataset title | Dataset URL | Database, license, and accessibility information |
|---|---|---|---|---|
| Blythe SA, Wieschaus EF | 2016 | ATAC-seq analysis of chromatin accessibility and nucleosome positioning in Drosophila melanogaster precellular blastoderm embryos | http://www.ncbi.nlm.nih.gov/geo/query/acc.cgi?acc=GSE83851 | Publicly available at the NCBI Gene Expression Omnibus (accession no: GSE83851) |

The following previously published datasets were used:

| Author(s) | Year | Dataset title | Dataset URL | Database, license, and accessibility information |
|---|---|---|---|---|
| Harrison MM, Li X, Kaplan T, Botchan MR, Eisen MB, 2011 | 2011 | Zelda binding in the early Drosophila melanogaster embryo marks regions subsequently activated at the maternal-to-zygotic transition | http://www.ncbi.nlm.nih.gov/geo/query/acc.cgi?acc=GSE30757 | Publicly available at the NCBI Gene Expression Omnibus (accession no: GSE30757) |
| Negre N, Brown CD, Morrison CA, Shah PK, White KP | 2009 | Genomewide analysis of Insulator protein binding sites in Drosophila embryos at E0-12 | http://www.ncbi.nlm.nih.gov/geo/query/acc.cgi?acc=GSE16245 | Publicly available at the NCBI Gene Expression Omnibus (accession no: GSE16245) |
| modEncode Consortium | 2009 | Drosophila at different time points of development: ChIP-chip, ChIP-seq, RNA-seq | https://www.ncbi.nlm.nih.gov/geo/query/acc.cgi?acc=GSE15292 | Publicly available at the NCBI Gene Expression Omnibus (accession no: GSE15292) |

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
