## [Decision Letter]

Thank you for submitting your article "Establishment and maintenance of heritable chromatin structure during early *Drosophila* embryogenesis" for consideration by *eLife*. Your article has been favorably evaluated by Kevin Struhl (Senior Editor) and three reviewers, one of whom is a member of our Board of Reviewing Editors. Two of the three reviewers have agreed to reveal their identity: Michael Eisen (Reviewer #2) and Ken Zaret (Reviewer #3).

The reviewers have discussed the reviews with one another and the Reviewing Editor has drafted this decision to help you prepare a revised submission.

Summary:

This is a very nice study, that was thoughtfully and carefully executed. The authors used ATAC-seq to probe the chromatin structure of individual pre-blastoderm *Drosophila* embryos during the final three cleavage divisions leading up to the mid-blastula transition (MBT), a fundamental transition for developing embryos that is of wide general interest. Major chromatin changes leading up to MBT are expected from previous work in *Drosophila* documenting cytological heterochromatin formation during this period. Employing staged individual embryos allows an exceptionally high and unprecedented level of temporal and cell cycle resolution. Consequently, the study is able to address widely interesting and important issues, including how chromatin accessibility changes during gene activation, and how chromatin is affected by rapid DNA replication and short cell cycle times. While the results are mostly descriptive, they strongly support particular mechanisms and provide a new, higher-level framework for all future studies of this period of embryonic development. Overall, the work represents an important step forward in our understanding of chromatin changes in the early embryo, and it will have a big influence on the field.

Essential revisions:

1) Greater clarity of presentation for a broad audience. The manuscript was well written but in a short and highly compressed format. Consequently, multiple intellectual and technical issues are handled in a user-unfriendly manner, apparently to save space. Although carrying out new experiments is not necessary, the authors should take advantage of *eLife*'s flexible format to make changes that will render the presentation more accessible to a broad audience. Instead of leaving most discussion to the end of the paper, the manuscript would become more accessible if critical questions of interest were raised first, then the rationale of how the experiments and bioinformatic processing can test the issues was presented, all before describing the results and conclusions.

For example, the ATAC-seq method underlies the entire paper and its strengths and limitations should be discussed in an introductory section. Based on their experiments, and previous knowledge, authors should discuss whether ATAC-seq "openness" is mostly a reflection of nucleosome positioning along the primary chromatin fiber. Is there any sensitivity to "higher order" structure? If so, it is not currently apparent and should be documented. Whether the binding of regulatory proteins can be detected should be addressed, as transcription factor binding is not currently apparent given that enhancers become more open in association with gene activation.

It is essential to present real numerical data for the y-axes of Figure 1 that are comparable for all the tracks shown and transparently reveal all data manipulations that were used. Currently, it is not fully clear how the presented data could be reproduced, since there is vague mention of normalization steps, discarding of data and the occasional use of two rather than one embryo (do you divide by 2?).

As an example of user unfriendly data analysis, Figure 3 plots "mean-scaled heterogeneity in PCNA-EGFP". There is no explanation why "mean-scaled heterogeneity" represents a logical way to summarize DNA replication during the cell cycle (or even what it means). The majority of readers will remain mystified. Several of the authors' later arguments require that DNA replication take place in a highly front-loaded manner during S phase. That way, reductions in nucleosome positioning that persist only during the start of S can be ascribed to DNA replication interference. However, a case that PCNA-EGFP heterogeneity provides such information is not made.

One of the most interesting and unique aspects of these data are their implication that rapid replication shortly after M phase represents a significant impediment to establishing/maintaining nucleosome positions and by inference, gene regulation. The authors should raise the issue of interference between replication and nucleosome positioning early in the paper, during analysis of Figure 1 data. Existing ATAC-seq peaks should be plotted and shown to double across the cell cycle and fall after M. Discrepancies (such more than a twofold decrease in early S phase) should be quantitated numerically and in duration to provide a quantitative basis for discussing interference. The paper should tout the analysis of NFR both as a mark of promoter opening, as well as a tool for analyzing replication effects and better explain the data processing, modeling and conversion to "relative NFR." Finally, the authors should explain why replication interference would be manifested as a cell cycle dependent oscillation in relative NFR, rather than simply as an oscillation in NFR signal strength without a size change due to essentially random nucleosome removal.

2) Too few primary data are shown to appropriately document these findings and address closely related questions. Although carrying out new experiments is not necessary, sample data at NC11-13 from other genomic regions containing well known genes activated in early embryos should be shown to allow the behavior of more peak regions to be observed. These should include a large gene such as Ubx which has been claimed to be too large to transcribe fully prior to MBT. Can ATAC-seq detect heterochromatin formation during NC11-13 in unique sequence centric regions? Including any available data on NC9 and NC10 embryo in Figure 1, even if of lower resolution, would help reveal more about the onset of the 33% of peaks already present at NC11.

3) More biological analysis to enhance the paper's impact for the general community. An important part of the study concerns the grouping of open region peaks into functional categories: promoters, enhancers, insulators, etc. The accuracy and biological meaning of these groupings is not currently presented in a critical or convincing manner. First, enhancers and insulators were identified using ChIP-seq data from much older embryos. What is the relevance of 0-12hr embryo data to NC11-NC13? The definitions used for insulators and enhancers in terms of protein binding and/or chromatin modifications seemed questionable and were not discussed or justified. The authors should prepare a (smaller) dataset of elements that have been functionally validated during NC11-13 and the next two hours of development and determine if they are consistent with conclusions based on their current dataset. The conclusion that (on average) enhancers become open before promoters, is touted in the Abstract, but its meaning and significance are scarcely discussed. Do such conclusions about opening of promoters and enhancers hold up using groups of specific known genes that become active at different times? When do the enhancers and promoters of gap and pair-rule genes become open, for example? How does the observed landscape of chromatin changes relate to what is known about gene action in early embryos?

4) Address issues regarding the maintenance of chromatin states through mitosis. There are three issues here: 1) whether open configurations are maintained through M, 2) whether maintenance is due to an epigenetically based "transcriptional memory", and 3) whether maintenance has an actual function- namely to ensure faithful inheritance. The fact open configurations survive M seems clear, since embryos in M phase display no loss of open features. However, is it possible, due to changes in embryonic chromatin during sample preparation, that apparent maintenance might be an artifact of partial progression of some processes into the next cell cycle? Since changing levels of trans acting factors likely cause these chromatin features in the first place, the re-appearance of features after M phase is to be expected whether or not there is any epigenetic "transcriptional memory." None of the data presented by the authors currently presents a convincing case that nucleosome positioning is inherited "epigenetically" and beyond that there were no functional tests of the importance of such inheritance. What in this study goes beyond the work of Hsuing and Blobel published this last year? Ramachandran and Henikoff 2016 should be cited and be clear about their own novelty here.

5) Address issues regarding the analysis of elements in haploid vs diploid embryos. Haploid embryos are mutant, and may not be entirely normal. The authors describe how many elements responded to N/C ratio (defined by the haploid/diploid test) or not, but the biological importance of this distinction is not well justified. The authors describe how many peaks overlap with binding sites for Zelda or for GAF (not necessarily in NC11-13 embryo cells), and find correlations with N/C dependent or independent classes. While this is consistent with a model in which GAGA factor plays a role in N/C ratio sensing, it is far from proof of such a relationship. It would be useful to see if there is genomic or other data out there on the genes targeted by Zelda and GAF factors in CHIP or knockout studies around the MBT and if the expected genes of the authors are affected.

---

## [Author Response]

[…]

*Essential revisions:*

*1) Greater clarity of presentation for a broad audience. The manuscript was well written but in a short and highly compressed format. Consequently, multiple intellectual and technical issues are handled in a user-unfriendly manner, apparently to save space. Although carrying out new experiments is not necessary, the authors should take advantage of eLife's flexible format to make changes that will render the presentation more accessible to a broad audience. Instead of leaving most discussion to the end of the paper, the manuscript would become more accessible if critical questions of interest were raised first, then the rationale of how the experiments and bioinformatic processing can test the issues was presented, all before describing the results and conclusions.*

We have revised the manuscript to clarify the rationale, approach, and experimental results with a more general reader in mind. We have added additional introductory information to the manuscript that presents the critical questions we are addressing. See also: further responses below.

*For example, the ATAC-seq method underlies the entire paper and its strengths and limitations should be discussed in an introductory section. Based on their experiments, and previous knowledge, authors should discuss whether ATAC-seq "openness" is mostly a reflection of nucleosome positioning along the primary chromatin fiber. Is there any sensitivity to "higher order" structure? If so, it is not currently apparent and should be documented. Whether the binding of regulatory proteins can be detected should be addressed, as transcription factor binding is not currently apparent given that enhancers become more open in association with gene activation.*

We have added a passage to the Introduction describing ATAC seq, its limitations, and the biological interpretation of ‘openness’ as measured by the assay (Introduction paragraph 4). Because ATAC-seq is a relatively new technology, some of the likely technical limitations have not been fully resolved in the literature. We have therefore approached the possibility of technical limitations empirically. The major “higher order structure” that could be problematic in our study is the condensed state of mitotic chromatin. We have added a section to the Materials and methods (“Read Length Distribution and Estimation of Stage-Specific Library Preparation Bias”) that outlines in detail how we approached the issue of comparing interphase and mitotic chromatin samples. This includes an additional supplemental figure (Figure 1—figure supplement 3) with raw read length distribution data and estimates of initial fragmented DNA concentration and actual unique sequence read distributions from interphase and metaphase staged samples (raw data available in [Supplementary-material SD2-data]). We do observe the expected change in relative proportions of nucleosome associated and open chromatin fragments depending on cell cycle stage. However, we find no inherent overall bias in the library preparation from metaphase-staged embryos, and therefore conclude that the data accurately reflect the accessibility status of metaphase chromatin.

With respect to regulatory protein binding, this phenomenon of ‘footprinting’ originally observed in Buenrostro 2013 can be seen with really high sequencing depth, beyond – we assume – what we have done here. We have added a passage to the Materials and methods (under the “ATAC-seq Sample Preparation” section) that mentions this. Practically, we had the choice to sequence many timepoints to good coverage, or to sequence fewer samples to very deep coverage. We chose to resolve time in this case. As we mention in the added text, we did not make any effort to determine whether, at the current sequencing depth, we nonetheless observe this TF footprinting.

*It is essential to present real numerical data for the y-axes of Figure 1 that are comparable for all the tracks shown and transparently reveal all data manipulations that were used. Currently, it is not fully clear how the presented data could be reproduced, since there is vague mention of normalization steps, discarding of data and the occasional use of two rather than one embryo (do you divide by 2?).*

The y-axes for the coverage plots in Figure 1 are now shown in the revised version.

We have now included an R markdown document that has annotated code for performing normalization of the data. The normalization approach described in the new code supplement effectively follows step by step the initial description of data normalization we included in the Materials and methods. In principle, a reader could prepare input datasets (either their own replicates of this data analysis, or their own homemade ATAC-seq data) and change a few lines of code in the markdown, and perform the same normalization as is used in the paper.

The “discarding” of data mentioned above refers not to discarding of sequenced libraries (i.e., cherry-picking), but rather discarding of libraries as a result of quality control assessments made during preparation (see end of fourth paragraph of Materials and methods, subheading “ATAC-seq Sample Preparation”). To clarify, all that is meant by ‘discarding’ here is that, following preparation of all libraries for a particular genotype, we estimated the starting quantity of fragmented DNA from a single sample by dividing the final library concentration by 2^(number of PCR cycles). The distribution of these values were plotted and outliers (3x the interquartile range) were discarded. The rationale for this is that, when preparing libraries from a single embryo, we were concerned that either some of the nuclear pellet would be lost during the initial sample preparation, or that the sample would be contaminated by exogenous DNA from the environment (e.g., contamination of embryo preps with brewer’s yeast from the embryo collection plates). If any of these were the case, then we would expect to resolve these as extreme outliers less than or greater than the population mean. In practice, these outliers were rare and constituted <5% of all samples prepared for this study. Those that we did find represented samples where we suspected we lost some or all of the nuclear pellet prior to Tn5 fragmentation.

We did not divide by two when we used two embryos. In general, we found that the quantity of DNA in the libraries did not vary significantly with respect to stage or DNA content (the expected 4-fold difference per embryo in genomic DNA content across all stages tested). This can be attributed to the large quantity of mitochondrial DNA that is also present in the library preparations. We have added a new paragraph to the Materials and methods, under the “ATAC-seq Sample Preparation” subheading, that describes the ‘buffering’ role of mitochondrial DNA in the library preparation steps.

*As an example of user unfriendly data analysis, Figure 3 plots "mean-scaled heterogeneity in PCNA-EGFP". There is no explanation why "mean-scaled heterogeneity" represents a logical way to summarize DNA replication during the cell cycle (or even what it means). The majority of readers will remain mystified. Several of the authors' later arguments require that DNA replication take place in a highly front-loaded manner during S phase. That way, reductions in nucleosome positioning that persist only during the start of S can be ascribed to DNA replication interference. However, a case that PCNA-EGFP heterogeneity provides such information is not made.*

We agree that this was confusing in the manuscript we initially submitted, and we have made the following changes to clarify this analysis.

We have added a supplemental figure attached to Figure 3 that illustrates directly what we are measuring, and have added additional clarification of this rationale to the Materials and methods. PCNA forms bright foci in syncytial blastoderm embryos at sites of DNA replication. The supplemental figure demonstrates representative frames from one of the movies that we used in this analysis, and these frames are matched to individual points along a plot of “heterogeneity” (1-Homogeneity) to help the reader interpret the plotted data. McCleland et al., JCB 2009 showed nicely that these patterns of heterogeneous fluorescence intensity are directly linked to DNA replication activity, insofar as blocking DNA origin licensing effectively eliminates the formation of these foci. We also note in the Materials and methods that a recent report has also approached this problem to the same end by quantifying the coefficient of variation in PCNA fluorescence (Deneke et al., Dev.Cell 2016). Follow-up work from the O’Farrell group (Shermoen, et al. Curr.Biol 2010) has demonstrated that replication during NC11 and NC12 does not largely distinguish between early and late replicating domains that are typically ascribed to eu- and hetero-chromatic compartments, but that during NC13 late replicating domains partly begin to emerge. Since we exclusively consider “euchromatic” portions of the genome in this manuscript, we are confident that our PCNA measurements provide a useful estimate of when DNA replication could be interfering with the underlying organization of chromatin in these regions. The ‘front-loaded’ aspect of replication in early embryos is a long-standing observation, which is consistent with our observations here. By measuring frequencies of observed ‘eye-forms’ of replication forks in EM-spreads of blastoderm DNA, Blumenthal et al. 1974 estimated that the interval of time between initiation and completion of replication is approximately 1.25 minutes, and that the vast majority of interphase genomic chromatin is post-replicative. See Figure 5 and text on pages 209-210 in Blumenthal for their argument. This is consistent with a model where DNA replication during the blastoderm cell cycle is highly front-loaded.

*One of the most interesting and unique aspects of these data are their implication that rapid replication shortly after M phase represents a significant impediment to establishing/maintaining nucleosome positions and by inference, gene regulation. The authors should raise the issue of interference between replication and nucleosome positioning early in the paper, during analysis of Figure 1 data. Existing ATAC-seq peaks should be plotted and shown to double across the cell cycle and fall after M. Discrepancies (such more than a twofold decrease in early S phase) should be quantitated numerically and in duration to provide a quantitative basis for discussing interference. The paper should tout the analysis of NFR both as a mark of promoter opening, as well as a tool for analyzing replication effects and better explain the data processing, modeling and conversion to "relative NFR." Finally, the authors should explain why replication interference would be manifested as a cell cycle dependent oscillation in relative NFR, rather than simply as an oscillation in NFR signal strength without a size change due to essentially random nucleosome removal.*

We considered rearranging the paper to include a more in-depth presentation of the replication/nucleosome question earlier in the Results section and chose instead to introduce the problem more generally in the Introduction (Paragraph 2 of the Introduction now covers the general issue of both replication and mitosis with respect to maintenance of accessibility states). In the current organization of the manuscript, we do in fact use the Figure 1 data to introduce our specific treatment of the replication/nucleosome question, albeit later in the Results section than the first mention of Figure 1. The reason for this is that we organized the paper to first deal with ‘establishment’ and then ‘maintenance’ because we felt this provided a more intuitive flow to the logic of the paper. We hope that this is a satisfactory compromise.

We have touted the usefulness of the NFR as an analysis tool in the revised Results section.

We have added additional technical detail for the calculation of the NFR sizes to the Materials and methods under the “Nucleosome Positioning” subheading. We also corrected some mis-numbered references to figures in this section.

The reason that replication associated nucleosome disruption is manifested as an oscillation in NFR size is related to the way our calculation of NFR size reacts to the effect of nucleosome placement following replication. From a single vantage point (the TSS) we are calculating the position of the first upstream and first downstream nucleosome, not the overall signal intensity within the maximum limits of the NFR region. This distinction, we think, addresses the heart of the request above. When nucleosome insertion occurs following replication, there is a chance that the insertion will occur within the NFR, in which case, our calculation indicates a reduction in NFR size. Alternatively, to invoke the Ramachandran and Henikoff model, if sufficient competition exists between regulatory factors and nucleosome assembly mechanisms, then it is less likely that a nucleosome will be inserted within the NFR (or that it will be captured in our measurements). In this case, our calculation indicates less of an effect on NFR size.

To address this in another way, we have now also measured changes in recovery of ATAC-seq fragments over the NFR as requested by the reviewer. To do this, we determined the maximum extent of the NFR by calculating the extreme positions of the -1 and +1 nucleosomes as predicted by our analysis. Then, we calculated the maximum ‘open’ ATAC-seq coverage value within each NFR. Accessibility increases by approximately two fold over each of the cell cycles measured, and there is less than a two-fold decrease in accessibility from mitosis into the subsequent early S-phase. We also measured how long it takes during either NC12 or NC13 for accessibility to recover to levels greater than or equal to accessibility during the preceding metaphase. On average, recovery of accessibility to prior metaphase levels occurs by 9 minutes into NC12, but by 6 minutes into NC13. We have added this observation to the Results text and have plotted the data in Figure 4—figure supplement 2.

We are grateful for the suggestion to pursue this analysis, as we became intrigued by the distribution of recovery times for NFRs. Examination of the distribution of this data indicated that the distribution of recovery times is biphasic in both cell cycles. To visualize this, we performed spline interpolation of the accessibility values for each NFR, and repeated the calculation of recovery times. During each cell cycle, there is an early (3-minute) and late (6-9 minute) population of recovering NFRs. The proportion of early and late NFRs is nearly equal during NC12, whereas there are a substantially greater proportion of early-recovering NFRs during NC13. These observations support our prior conclusion that recovery of chromatin architecture at NFRs becomes more robust during NC13. This new analysis is summarized as well in Figure 4—figure supplement 2.

*2) Too few primary data are shown to appropriately document these findings and address closely related questions. Although carrying out new experiments is not necessary, sample data at NC11-13 from other genomic regions containing well known genes activated in early embryos should be shown to allow the behavior of more peak regions to be observed. These should include a large gene such as Ubx which has been claimed to be too large to transcribe fully prior to MBT. Can ATAC-seq detect heterochromatin formation during NC11-13 in unique sequence centric regions? Including any available data on NC9 and NC10 embryo in Figure 1, even if of lower resolution, would help reveal more about the onset of the 33% of peaks already present at NC11.*

We have added a selection of 23 plots (including Ubx) similar to the ones shown in Figure 1 as supplemental figures to Figure 1. These are most but not all of the genes referred to in the revised text. In reality, it would be impractical to show a nice large set of example regions that would cover all the early embryo favorites. The necessary files for making these plots are available through GEO, and we have posted a link on the lab webpage to allow readers to browse through the data through the UCSC genome browser (http://molbiolabs.princeton.edu/wieschaus/data). If there are even more ways to disseminate these data files beyond what we have done here, we would be interested in pursuing those as well. We encourage the referees to examine the link above and comment on the presentation of the data. We note in both the Figure 1 legend and in the Materials and methods the options available to the reader for viewing their own favorite loci.

We are likewise interested in the question of heterochromatin formation in early embryos, but are not satisfied that we have made any significant progress on this question. If we take the centric regions of chromosomes 2, 3, and X, mask them for regions that are uniquely mappable, and calculate the average ATAC-seq ‘open’ coverage over these regions, we find that even at NC11, mean heterochromatic accessibility is lower than that of an equal number of randomly sampled euchromatic regions. In line with the prediction that cytological heterochromatin emerges at or around the MBT, we observe that NC12 and NC13 have progressively less open heterochromatin. These differences between eu- and hetero-chromatin are statistically significant by permutation test at p<0.001 for all timepoints in the analysis, and we have not tested whether the progressive reductions we observe from NC11 to 13 are themselves statistically significant. In this latter case, we are unsure of the significance of the magnitude of the observed effect, and would overall prefer to link these measurements to functional assays that are currently in development. This initial analysis is promising, but we are not ready to include this as a major observation associated with the current study.

We are also interested in evaluating accessibility in earlier embryos, but have not yet performed such experiments. We agree that this would help resolve the NC11 peaks into more distinct classes. One reason that we did not push earlier in this analysis is that NC11 is the earliest developmental stage where we can confidently ascribe a timepoint within a nuclear cycle. Mitosis 9 occurs below the cortical layer of the embryo and therefore the beginning of NC10 is therefore difficult to observe. As a result, such earlier timepoints will indeed represent data of lower time resolution. This, coupled with the fact that we’d likely have to pool several earlier embryos of various stages, led us to the conclusion that such experiments would not necessarily fit with the current dataset from a sampling standpoint.

*3) More biological analysis to enhance the paper's impact for the general community. An important part of the study concerns the grouping of open region peaks into functional categories: promoters, enhancers, insulators, etc. The accuracy and biological meaning of these groupings is not currently presented in a critical or convincing manner. First, enhancers and insulators were identified using ChIP-seq data from much older embryos. What is the relevance of 0-12hr embryo data to NC11-NC13? The definitions used for insulators and enhancers in terms of protein binding and/or chromatin modifications seemed questionable and were not discussed or justified. The authors should prepare a (smaller) dataset of elements that have been functionally validated during NC11-13 and the next two hours of development and determine if they are consistent with conclusions based on their current dataset. The conclusion that (on average) enhancers become open before promoters, is touted in the Abstract, but its meaning and significance are scarcely discussed. Do such conclusions about opening of promoters and enhancers hold up using groups of specific known genes that become active at different times? When do the enhancers and promoters of gap and pair-rule genes become open, for example? How does the observed landscape of chromatin changes relate to what is known about gene action in early embryos?*

We have added additional text to the Materials and methods to justify the categorization of the enhancer class, which derives largely from previous observations of chromatin modifications and protein occupancy from a broad collection of genomics studies.

We would like to point out that for enhancers, we used a combination of functionally validated enhancers from the Stark Lab (Kvon et al., Nature 2014), binding of Zelda (Eisen Lab, 1-3h embryos), binding of CBP (modEncode 0-4h embryos), Histone H3K4 monomethylation (modEncode, 0-4h), and Histone H3K27 acetylation (modEncode, 0-4h). All of these data sets are as closely matched temporally as possible to the stages we are examining in the paper.

To validate the core conclusion we derive from the categorization of enhancers, we added the suggested analysis on the Kvon/Stark set of functionally validated enhancers to the Results and Discussion section and Figure 1. In brief, this subset of enhancers demonstrates the preferential early accessibility we observe with the broader set of ‘enhancers’. We note here that this collection does not include several well-characterized early enhancers (e.g., hunchback P2) and is therefore likely to be under-respresented for features in the early time class.

In the process of examining the validated enhancers, we also measured when these enhancers are active, based on the annotations supplied by Kvon/Stark. First, as expected, compared with the total set of Kvon/Stark enhancers our NC11-NC13 accessible regions are enriched for enhancers with the earliest scored expression (stage 4-6, where NC11-13 is during stage 4 and stage 6 is approximately 1.5 hours later, during gastrulation). Second, the enhancers that are accessible between NC11 and NC13 continue to be expressed throughout development (through stage 16, approximately ≥12 hours later). Third, enhancers that are open early (NC11) are more likely to be expressed early, whereas enhancers that open between NC12 and 13 are significantly enriched for enhancers whose first expression comes much later in development, suggesting that these enhancers are ‘primed’ during NC12 and 13 for expression at a later stage. Taken together, these observations support the conclusions we draw from the broader categorization of ‘enhancers’, and further suggest routes for future investigation regarding the phenomenon of enhancer priming.

With respect to our use of 0-12 hour ChIP-chip data for assigning insulators: we acknowledge that this data set is not ideal, but it is the most comprehensive available dataset that measures multiple insulator binding proteins, and we feel that even with this caveat, our analysis responsibly uses this data for categorization of this genomic feature. Our options for this analysis were 1) to use the 0-12 hour data; 2) generate all of this data ourselves for our stages of interest; 3) use a 2-4 hour dataset that lacks one of the major insulator proteins mod(mdg4); 4) not score insulators at all. Formally, we designate a region in our data that has measured accessible chromatin between NC11 and NC13 as an insulator if it ‘can’ bind two or more of the insulator proteins between 0 and 12 hours in the modEncode dataset. Given that these insulator proteins are expressed between NC11 and 13, it seems reasonable to assume that they would bind these regions once they were made accessible. Furthermore, our analysis with the Kvon/Stark enhancer set described above does suggest that events in NC11-NC13 do have lasting effects until much later in development. A more careful and direct analysis will therefore be necessary to truly answer what similarities there are between a 12 hour embryo and a syncytial blastoderm, but from the point of view of assigning genomic features, we think that this approach is the best available given the current state of the field.

Finally, we agree that it is important to link these observations back to real knowledge about gene activity and function in early embryogenesis. Similar to our response to the major comment above regarding providing more examples of primary data for more genes of interest, it would be difficult to cover all the favorites here. We have therefore added more general discussion regarding groups of certain favorite/developmentally rich genes to the Results and Discussion, and have provided a new supplemental table that summarizes the set of peaks corresponding to promoters, identifies them by associated flybase gene identifiers and names, and provides the scored “open by” category, and the NC-ratio independent or dependent classification, when appropriate. With this information, the general behavior of any reader’s favorite gene can be easily looked up, and coupled with the provided data visualization either on our website or by downloading the processed data, further investigation can be made into the finer details. For the majority of the genes we have discussed in this new section, we have included browser views in the Supplement to Figure 1, as described above.

*4) Address issues regarding the maintenance of chromatin states through mitosis. There are three issues here: 1) whether open configurations are maintained through M, 2) whether maintenance is due to an epigenetically based "transcriptional memory", and 3) whether maintenance has an actual function- namely to ensure faithful inheritance. The fact open configurations survive M seems clear, since embryos in M phase display no loss of open features. However, is it possible, due to changes in embryonic chromatin during sample preparation, that apparent maintenance might be an artifact of partial progression of some processes into the next cell cycle? Since changing levels of trans acting factors likely cause these chromatin features in the first place, the re-appearance of features after M phase is to be expected whether or not there is any epigenetic "transcriptional memory." None of the data presented by the authors currently presents a convincing case that nucleosome positioning is inherited "epigenetically" and beyond that there were no functional tests of the importance of such inheritance. What in this study goes beyond the work of Hsuing and Blobel published this last year? Ramachandran and Henik_off_ 2016 should be cited and be clear about their own novelty here.*

From a logical standpoint, we cannot unequivocally exclude the possibility that sample preparation somehow introduces an artifact that allows for the ‘partial progression of some processes into the next cell cycle’. However, we feel that this is unlikely. Another way to state the problem is whether our observation of maintained accessibility during metaphase is indeed real. We initially approached this observation with healthy skepticism along the lines of the abovementioned artifact. However, we were encouraged when we found Hsiung and Blobel’s recent Genome Research paper, which reports a similar maintenance of DNase hypersensitive sites in a murine erythroblast cell line during mitosis. Given the importance and relevance of Hsiung and Ramachandran’s previous studies, we have cited them more prominently in the manuscript, and have more clearly discussed how we advance their important prior observations.

Hsiung’s observations are made in a completely unrelated experimental model. In addition, Hsiung uses a completely different assay system, DNase hypersensitivity, to measure accessibility, and their approach to collecting mitotically staged samples is done pharmacologically, i.e., nocodazole arrest. Finally, Hsiung fixes the samples with formaldehyde, which more than likely prevents further biological activity. Given the similarity between our observations despite the significant differences in experimental approach, we take our results as demonstrating preserved local accessibility patterns in an otherwise condensed mitotic chromatin state, especially given that any artifact would have to apply to both our and Blobel’s divergent experimental systems.

Moreover, the reviewers’ question implies that “epigenetic transcriptional memory” is distinct from the effects of “trans-acting factors”, that transcription factors do one thing, and there exists a separate mechanism for memory. This is not necessarily the case. We agree that these are interesting questions, but our failure to identify changes in accessibility during mitosis argues that physical structures are maintained and thus preclude our addressing more subtle epigenetic mechanisms that might maintain chromatin memory over longer periods. We do not know the mechanism responsible for the stability of nucleosome positioning/ chromatin accessibility during mitosis. It may be the most thermodynamically favorable way for the cell to deal with the task of chromatin condensation, or it may result from a previously unappreciated ability for trans-acting factors to remain associated with their DNA targets during mitosis. It may also reflect a separate mechanism for epigenetic memory. We also have not determined whether the loss and recovery of nucleosome stability following DNA replication is deleterious to the precision with which transcriptional reactivation takes place. All these are big questions we hope to address in future investigations.

*5) Address issues regarding the analysis of elements in haploid vs diploid embryos. Haploid embryos are mutant, and may not be entirely normal. The authors describe how many elements responded to N/C ratio (defined by the haploid/diploid test) or not, but the biological importance of this distinction is not well justified. The authors describe how many peaks overlap with binding sites for Zelda or for GAF (not necessarily in NC11-13 embryo cells), and find correlations with N/C dependent or independent classes. While this is consistent with a model in which GAGA factor plays a role in N/C ratio sensing, it is far from proof of such a relationship. It would be useful to see if there is genomic or other data out there on the genes targeted by Zelda and GAF factors in CHIP or knockout studies around the MBT and if the expected genes of the authors are affected.*

We have clarified the experimental rationale for examining haploids in the Results and Discussion section.

We have added a detailed description of sesame and have clearly acknowledged any potentially confounding features of the of the mutant phenotype in the Materials and methods (new section entitled “Haploid Embryos”).

We agree and acknowledge that the correlations between Zelda/GAF and regions of open chromatin are simply that, correlations, and not proof of N:C ratio sensing. While detailed information regarding the effects of Zelda loss of function on chromatin accessibility have been reported previously (Schulz 2015, Sun 2015), loss of function for GAF in embryos is difficult to achieve, and – at best – represents rare pleiotropic escapers from an earlier ovarian sterile phenotype (Bhat et al., 1996, Greenberg et al., 2001). Correspondingly, all of our attempts to generate GAF-null embryos have failed and we are currently pursuing several alternative approaches to definitively address the mechanistic relationship between GAGA and the MBT timers.

To further substantiate the correlation between Zelda and N:C-ratio independent chromatin regions, we obtained a previously published Faire-seq dataset identifying regions of chromatin that are dependent on Zelda for accessibility (Schulz, 2015) and performed statistical analysis for enrichment of Zelda-dependent loci and our different timing classes. We note that in Schulz et.al, the fraction of Zelda-bound regions (i.e., ChIP-peaks) is much larger than the set of identified regions that are absolutely dependent on Zelda for full chromatin accessibility, and therefore represent a small subpopulation of the Zelda binding regions both we and Schultz et al. have used to identify “Zelda targets”. Nonetheless, we observe that the Schultz-Zelda-dependent regions are likewise enriched for N:C ratio independent acquisition of chromatin accessibility. Performing the identical Fisher’s exact test we use in Figure 2, we find that Zelda-dependent regions are enriched for N:C ratio independence with a p-value of 0.001415 and a log2 odds ratio of 1.11.

We were concerned that the Schultz dataset could potentially be missing some Zelda-dependent regions owing to differences in sample collection and because of unappreciated technical differences between ATAC-seq and Faire-seq. We therefore double-checked these results using an unpublished Zelda ATAC-seq dataset that is currently part of an independent manuscript from our laboratory that is currently under review. These data were collected from embryos staged twelve minutes into NC14 (approximately 15 minutes older than the oldest embryo examined in this manuscript). Of note, our unpublished dataset identifies substantially more regions that depend on Zelda for chromatin accessibility, yet is still consistent with Schulz et al. in the fact that not all Zelda bound regions require Zelda for accessibility. When we perform the statistical analysis as described above, we also find that regions that require Zelda for chromatin accessibility are more likely to be N:C ratio independent with a p-value of 2.097x10^-08^, and a log2 odds ratio of 0.78. We feel that these supplemental tests strongly support the conclusion from our manuscript that on average the effect of Zelda’s interaction with chromatin is timed independently of the nucleo-cytoplasmic ratio.

We also double-checked the correlation of GAF binding with N:C ratio-dependence. As mentioned above, no chromatin accessibility data from GAF-null embryos exists, or is technically possible to obtain at the moment. Therefore, we obtained a 0-4 hour GAF ChIP-seq dataset from modEncode (accession# 3814) and tested for a correlation between binding and N:C ratio dependence. With the statistical analysis described above, we find that regions bound by GAF and not Zelda in this dataset are more likely to be NC ratio dependent with a p-value of 0.014 and a log2 odds ratio of 0.56. In the future, we hope to be able to address this issue more directly with more closely time-resolved GAF ChIP-seq data and hopefully loss of function accessibility data.

We note that we have included the details of these analyses here in our response, but have not incorporated them into the manuscript proper.